

# Aquatic sloths (*Thalassocnus*) from the Miocene of Chile and the evolution of marine mammal herbivory in the Pacific Ocean

Ana M. Valenzuela-Toro[1,2], Nicholas D. Pyenson[2], Jorge Velez-Juarbe[3] and Mario E. Suárez[4,5]

[1] Área Científica, Centro de Investigación y Avance de la Paleontología e Historia Natural de Atacama, CIAHN Atacama, Caldera, Chile
[2] Department of Paleobiology, National Museum of Natural History Smithsonian Institution, Washington DC, Washington DC, United States
[3] Department of Mammalogy, Natural History Museum of Los Angeles County, Los Angeles, California, United States
[4] Atacama Fósil Limitada, Caldera, Chile
[5] Laboratorio de Ontogenia y Filogenia, Departamento de Biología, Facultad de Ciencias, Universidad de Chile, Santiago, Chile

Corresponding author
Ana M. Valenzuela-Toro,
avalenzuela.toro@gmail.com

## ABSTRACT

The evolution of marine mammals in South America includes unique and extinct lineages found nowhere else in the world, such as the walrus-convergent whale *Odobenocetops* and multiple aquatic sloth species belonging to the genus *Thalassocnus*. Aquatic sloths have been collected from Mio-Pliocene marine deposits in Peru and Chile, and terrestrial deposits in Argentina. In Chile, these occurrences range over 800 km across several basins from the Bahía Inglesa, Coquimbo, and Horcón formations. Here we report aquatic sloth material belonging to the species *Thalassocnus natans* from a new locality, Norte Bahía Caldera from the Bahía Inglesa Formation in the Atacama Region. We find multiple lines of evidence to support a late Miocene age for this material, which consists of a nearly complete skeleton, including cranial and postcranial remains and associated mandible and postcranial elements that represent the most complete *Thalassocnus* specimens reported yet from Chile. Based on this finding, we review the stratigraphic origin and geologic age of *Thalassocnus* species from the western coast of South America and determine that remains from the Upper Pliocene of central Chile represent the youngest known record of this genus to date. Our review also suggests that overlapping stratigraphic ranges for type material of *T. antiquus* with both *T. natans* and *T. littoralis* weakens the proposed argument for *Thalassocnus* evolution through anagenesis succession. Finally, in the context of *Thalassocnus* material from Chile and Peru, including other marine mammal herbivores (*e.g.*, sirenians), we demonstrate that one of the most unusual features of this guild is that South American marine mammal herbivores did not reach the body sizes of their analogous counterparts (*i.e.*, desmostylians and sirenians) in the North Pacific. This observation builds on other paleontological evidence about the unusual features of the Peruvian biotic province during the Neogene.

## INTRODUCTION

Aquatic sloths (*Thalassocnus* spp.) represent one of the most enigmatic cases of secondary adaptation to the marine environment among tetrapods. *Thalassocnus* is the only aquatic incursion among xenarthrans (sloths, anteaters, armadillos, and extinct relatives) and the the genus is placed within Megatherioidea, a clade of xenarthrans that includes giant ground sloths and other terrestrial lineages (*Tejada et al., 2024*; *Boscaini et al., 2025*). Five *Thalassocnus* species have been described from vertebrate-bearing marine sedimentary units of coastal Peru with each species putatively belonging to a different geologic age: *T. antiquus* (~9–8 Ma), *T. natans* (~7 Ma), *T. littoralis* (~7–6 Ma), *T. carolomartini* (~6–5 Ma) and *T. yaucensis* (probably early Pliocene) (*Muizon & McDonald, 1995*; *McDonald & De Muizon, 2002*; *Muizon et al., 2003*, *2004a*; *Ochoa et al., 2021*).

The aquatic adaptations of *Thalassocnus* were originally based on taphonomic evidence (*i.e.*, preservation in marine sediments) and traits in the postcranial skeleton of *T. natans* (*Muizon & McDonald, 1995*). Further analyses supported an aquatic habit and indicated a gradual morphological adaptation to foraging on marine vegetation from the oldest to the youngest species of the genus (*Muizon et al., 2004b*). Specifically, it has been proposed that *T. natans*, *T. antiquus*, and *T. littoralis* were mixed feeders, likely consuming some degree of seagrasses in shallow waters, whereas *T. carolomartini* and *T. yaucensis* displayed a more specialized feeding morphology linked with grazing (*Muizon et al., 2004b*). The functional morphology of the appendicular skeleton suggests an adaptation to aquatic locomotion over geologic time, with an increased strength for hand gripping, likely enhancing underwater foraging (*Amson et al., 2015a*, *2015b*, *2015c*). Likewise, histological studies have demonstrated that *Thalassocnus* species displayed osteological modifications consistent with a gradual adaptation for buoyancy control, including the development of pachyostosis and osteosclerosis similar to sirenians (*Amson et al., 2014*; *Amson, Billet & de Muizon, 2018*; *Cooper & Maas, 2018*).

Remains of *Thalassocnus* spp. were initially considered restricted uniquely to the Pisco and Sacaco basins of southern Peru. However, additional material has been recovered from late Miocene and Pliocene marine fossiliferous deposits in northern and central Chile (*Canto et al., 2008*; *Pyenson et al., 2014*; *De Los Arcos et al., 2017*) (Fig. 1). More recently, *Thalassocnus* was reported from continental settings from the late Miocene-Pliocene Tafna Formation in the Jujuy Province on the Altiplano in Central Andes of northwestern Argentina (*Quiñones et al., 2022*). This surprising occurrence broadened the geographical distribution of this genus to around 1,000 km inland from the nearest coastal areas in the Central Andes (*Quiñones et al., 2022*), indicating a more complex paleobiogeographic and paleoecological history for this group.

In Chile, remains of *Thalassocnus* have been recovered from fossiliferous marine strata of the Bahía Inglesa Formation in the Atacama Desert (Table 1). They correspond to isolated cranial and postcranial elements and have been identified as belonging to

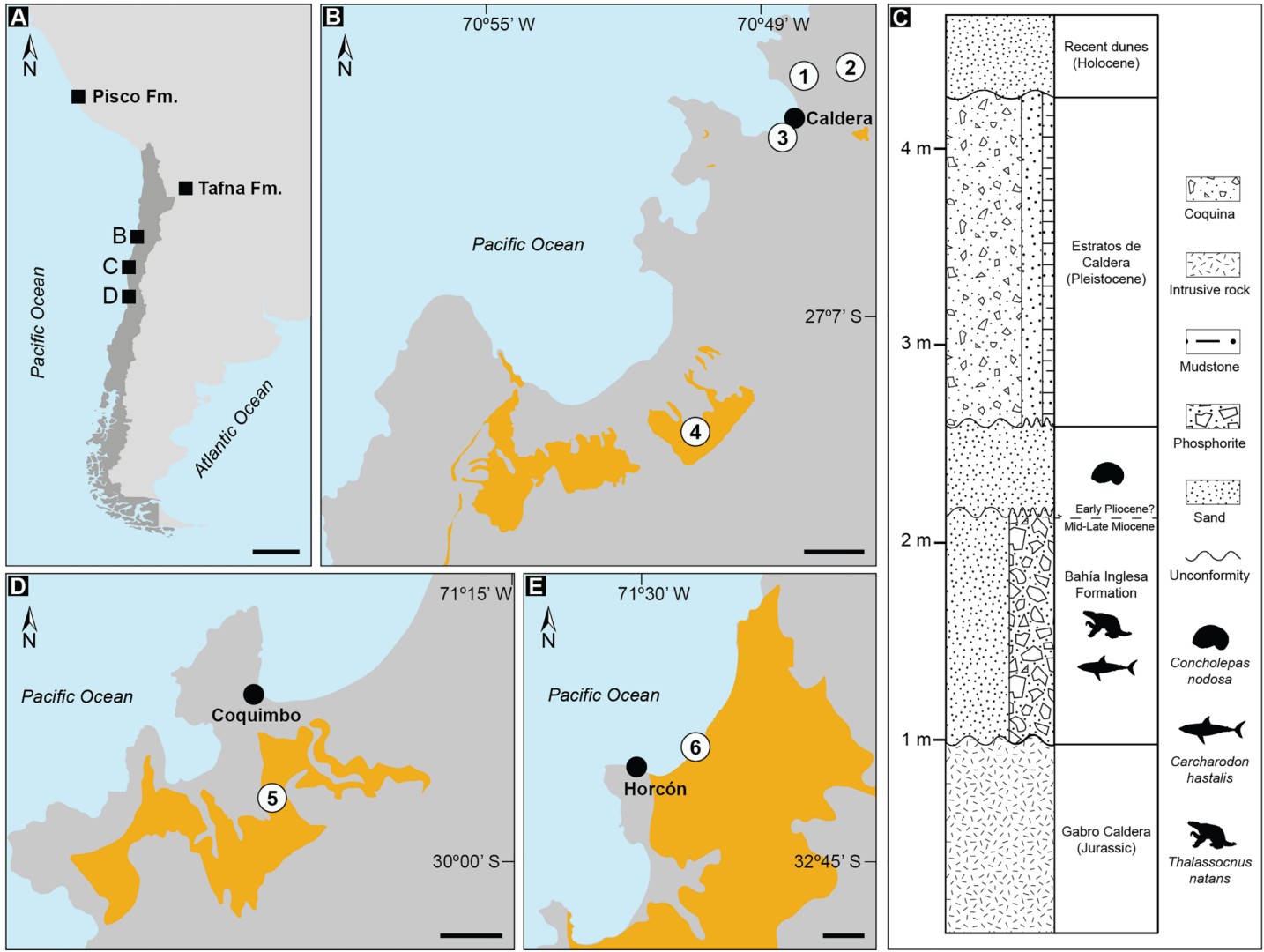

**Figure 1 Records of *Thalassocnus* from southwestern South America.** Geologic formations with published occurrences in Argentina (Tafna Fm.), Peru (Pisco Fm.), and Chile (A). Details of localities where aquatic sloth remains have been excavated from the Bahía Inglesa Formation (B), and the stratigraphic context of the fossil remains reported here from the locality Norte Bahía Caldera (C). Sites where *Thalassocnus* remains have been recovered include the Coquimbo (D) and Horcón (E) formations. Legends: 1. Norte Bahía Caldera; 2. Cerro Ballena; 3. Arenas de Caldera; 4. Parque Paleontológico Los Dedos (Mina Fosforita) [location inferred from *Peralta-Prato & Solórzano (2019)*]; 5. Lomas del Sauce; 6. Playa La Luna. Black scales represent 500 km in A and 2 km in (B, D, and E). Yellow outlines indicate fossiliferous outcrops within the respective formations. Animal silhouettes are not to scale and were obtained from Phylopic (phylopic.org).

*T. natans* and *Thalassocnus* sp. (*Canto et al., 2008*; *Pyenson et al., 2014*). In addition, associated and isolated postcranial remains have been reported from Pliocene fossil beds of the Coquimbo and Horcón formations in northern and central Chile (Fig. 1), which have been assigned to *T. carolomartini* and *T. natans*, respectively (*De Los Arcos et al., 2017*). These records show that the lack of associated skeletal material for *Thalassocnus* from Neogene Chilean localities prevents clear species-level assignments and hinders biostratigraphic comparisons with other South American occurrences of this taxon.

**Table 1 Published records of *Thalassocnus* from Chile.**

| Collection number | Original identification | Revised identification | Locality | Formation | Element | Reference |
|---|---|---|---|---|---|---|
| SGO.PV 1093 | *Thalassocnus* sp. (*T. antiquus* or *T. natans*) | *Thalassocnus* sp. | Estanques de Copec (now referred as Norte Bahía Caldera) | Bahía Inglesa | Partial right mandible with m1–m3 | *Canto et al. (2008)* |
| MPC-SPN2 (Current collection number: MPC 704-A) | *Thalassocnus natans*, *Thalassocnus* cf. *T. natans* | *Thalassocnus natans* | Norte Bahía Caldera | Bahía Inglesa | Postcranial skeleton and a cranium | *Suárez et al. (2011)*, *Velez-Juarbe et al. (2012)*; this study |
| MPC 705 | *Thalassocnus natans* | – | Norte Bahía Caldera | Bahía Inglesa | Isolated left dentary | This study |
| MPC 705-A | *Thalassocnus natans* | – | Norte Bahía Caldera | Bahía Inglesa | Associated right femur and a left tibia | This study |
| SGO.PV 1133 | *Thalassocnus* sp. (*T. natans* or *T. littoralis*) | *Thalassocnus* sp. | Arenas de Caldera | Bahía Inglesa | Two femora | *Suárez et al. (2011)* |
| MPC 704 | *Thalassocnus natans* | Folivora indet. | Cerro Ballena | Bahía Inglesa | Posterior part of left horizontal ramus of edentulous mandible | *Pyenson et al. (2014)*, *Amson et al. (2015b)*; this study |
| MPC 644 | *Thalassocnus natans* | *T. natans* | Cerro Ballena | Bahía Inglesa | Right femur | *Pyenson et al. (2014)*; this study |
| CPUC/C/557 | *Thalassocnus* cf. *T. natans* | – | Mina Fosforita member | Bahía Inglesa | Distal fragment of humerus | *Peralta-Prato & Solórzano (2019)* |
| SGO.PV 15500 | *Thalassocnus carolomartini* | – | Lomas del Sauce | Coquimbo | Partial postcranium | *De Los Arcos et al. (2017)* |
| SGO.PV 21545 | *Thalassocnus littoralis?, carolomartini?, yaucensis?* | *Thalassocnus* sp. | Playa la Luna | Horcón | Isolated right proximal and middle phalanges of the third digit of the pes | *De Los Arcos et al. (2017)* |

**Note:**
Institutional abbreviations: CPUC/C: Geological Museum "Lajos Biró Bagóczky," Universidad de Concepción, Concepción, Chile; SGO.PV: Museo Nacional de Historia Natural, Santiago, Chile; MPC: Museo Paleontológico de Caldera, Caldera, Chile.

Here, we report new fossil material, comprising a nearly complete skeleton, including cranial and postcranial remains, an isolated dentary, and associated postcranial elements (MPC 704-A, 705, 705-A) of *Thalassocnus* from the Late Miocene of the Bahía Inglesa Formation in the Atacama Desert. We discuss the implication of this occurrence for the fossil record of this genus in South America in the context of the evolution of marine mammal herbivory across the Pacific Ocean.

# MATERIALS AND METHODS

## Specimens observed

*Thalassocnus antiquus* (MUSM 228); *T. carolomartini* (MNHN SAS 201, 203); *T. littoralis* (MNHN SAS 40–42, 44, 45, 53, 56, 57, 61, 158, 256, 1611, 1620, 1621; MUSM 223); *T. natans* (MNHN SAS 734, MUSM 433); *T. yaucensis* (MUSM 37, 347, 434).

## Locality and geologic age

The new aquatic sloth specimens (MPC 704-A, MPC 705, and MPC 705-A) were collected in April 2011 by one of the authors (MS) from a new locality, Norte Bahía Caldera (NBC) located ~4 km north of the town Caldera and ~1.5 km southwest of Cerro Ballena in the

Atacama Region of Chile, at the coordinates 27°03′02.4″S, 70°48′23.9″W (Figs. 1A, 1B). In the Caldera Basin, the Neogene fossil-bearing rock units are represented by the Bahía Inglesa Formation (*Rojo, 1985*), which is abundantly exposed in the area, including at Cerro Ballena (*Pyenson et al., 2014*). Many of these exposures also include Estratos de Caldera, a Pleistocene marine terrace deposit capping the sequence (*Valenzuela-Toro et al., 2013*). Recently, *Martinez et al. (2025)* used chronostratigraphy to correlate exposures of the Bahía Inglesa Formation at Cerro Ballena with exposures from the same formation 15 km south at Mina Fosforita in the Parque Paleontológico Los Dedos based on U-Pb zircon ages.

There are several lines of evidence to suggest that NBC belongs to the Bahía Inglesa Formation. First, NBC is located about 6 m in elevation, whereas Cerro Ballena is about 60 m in elevation. There are no visible faults in the area north of Caldera, suggesting that the elevation difference is unfaulted (*Maldonado, Contreras & Melnick, 2021*) and that NBC is stratigraphically below Cerro Ballena. At NBC, Pleistocene Estratos de Caldera overlies a phosphatic green sandstone unit that is the source rock for the aquatic sloth material reported here. This lithology is unlike any of the units at Cerro Ballena, and is not reported from Bahía Inglesa Formation sections measured at Parque Paleontológico Los Dedos nor further south.

*Canto et al. (2008)* collected a mandible remain that they referred to *Thalassocnus* from Estanques de Copec, which they identified as "next to N° 5 national route" about 6 km northeast of Caldera at 27°02′S, 70°48′W. This description and these coordinates place *Canto et al.*'s *(2008)* locality ~1.5 km northeast of Cerro Ballena, to the west of the Pan-American highway and 2 km north of NBC, which is less than 500 m from Planta Copec Caldera, a facility today that is likely the name-bearer of *Canto et al.*'s *(2008)* Estanques de Copec and at the same elevation as NBC. Thus, we suggest that Estanques de Copec is likely synonymous or at the same stratigraphic horizon as NBC.

*Canto et al. (2008)* described their *Thalassocnus* material as belonging to the Lechero Member of the Bahía Inglesa Formation, following *Walsh & Hume*'s *(2001)* name for Unit 3 of the Bahía Inglesa Formation located above the main phosphatic hardground deposits (Unit 2) abundantly exposed at the locality Parque Paleontológico Los Dedos. Prior to *Martinez et al.*'s *(2025)* revision of the formation at Los Dedos, the age of the Lechero Member has been proposed as 4.5–2.6 Ma based on microfossil biostratigraphy evidence (*Ibaraki, 1995*; *Tsuchi et al., 1988*), but the geographic locality for these data remains unclear. Other authors (*e.g.*, *Long, 1993*, *Suárez & Marquardt, 2003*, *Bianucci et al., 2006*) cite the presence of shark species *Carcharodon carcharias* and *Prionace glauca* and the fossil delphinid genus *Hemisyntrachelus* as Pliocene age indicators for material collected in Unit 3 above Unit 2, but none of these references are tied to specimens with stratigraphic control.

Lastly, while *Martinez et al. (2025)* make no explicit nomenclatural recommendations about the three units of the Bahía Inglesa Formation, their chronostratigraphy work provides new insights on the age of the fossiliferous strata. Specifically, the bulk of fossil vertebrate-bearing units in the Bahía Inglesa Formation at Parque Paleontológico Los Dedos are Late Miocene in age, with the lowest level of fossiliferous phosphatic sandstones

dating to 7.38 ± 0.53 Ma, while the overlying volcanic ash yielded a date of 5.50 ± 0.03 Ma, and the uppermost diatomite sequence was dated to 4.35 ± 0.07 Ma, with the latter two corresponding to the upper and lower ages for Unit 3.

The fossil assemblage at the NBC locality can be correlated with other previously reported sites within the Bahía Inglesa Formation, including Arenas de Caldera (*Suárez, Lamilla & Marquardt, 2004*) and Parque Paleontológico Los Dedos (*Pyenson et al., 2014*; *Valenzuela-Toro et al., 2013*). The fauna at NBC indicates the presence of two distinct stratigraphic sequences: a lower and an upper layer, likely Miocene and Pliocene in age, respectively. This age determination depends on biostratigraphic marker species for Neogene deposits in the southeastern Pacific. The lower sequence is a phosphatic level containing remains of the shark *Carcharodon hastalis* in association with specimens of *Thalassocnus* (Fig. 1C). This faunal assemblage points to a Middle to Late Miocene age for the unit (*Suárez & Marquardt, 2003*; *Ehret et al., 2012*; *Pyenson et al., 2014*). In the overlying sand levels, fossils of the muricid gastropod *Concholepas nodosa* were found, indicating a Pliocene age for this upper sequence (*Cárdenas, Viard & Castilla, 2008*). Overall, the geographic position, lithology, and biostratigraphic composition of the sequence at NBC is most consistent with those features of the Bahía Inglesa Formation.

## RESULTS

### Systematic Paleontology

Mammalia Linnaeus, 1758
Xenarthra Cope, 1889
Folivora Delsuc, Catzeflis, Stanhope, and Douzery, 2001
Megatheriidae Owen, 1843
Thalassocninae Muizon, McDonald, Salas, and Urbina, 2004
*Thalassocnus Muizon & McDonald, 1995*
*Thalassocnus natans Muizon & McDonald, 1995*

**Referred Specimens**—MPC 704-A, a semi-articulated skeleton consisting of an incomplete cranium, left radius, left ulna, elements of the left and right manus, left pelvis, right femur, left tibia, right calcaneus, right astragalus, right pedal metatarsal V and right digit 1–2 of the second pedal phalanx, several vertebrae, and fragmentary ribs (not described). We also report MPC 705, an isolated left dentary, missing angle and posterodorsal part of ascending ramus; and MPC 705-A, an associated right femur and a left tibia. All these remains were collected from the locality Norte Bahía Caldera (= "Estanques de Copec" in *Canto et al. (2008)*), Bahía Inglesa Formation, northern Chile. These specimens were referred to in previous publications under the temporary collection number MPC- SPN2 (*Suárez et al., 2011*; *Velez-Juarbe et al., 2012*; *De Los Arcos et al., 2017*; *Peralta-Prato & Solórzano, 2019*).

### Description and comparisons

**Skull**—MPC 704-A preserves a nearly complete skull (Figs. 2–4). The anterior palatal portion of the maxilla is complete on the right side, and it likely contacts with the

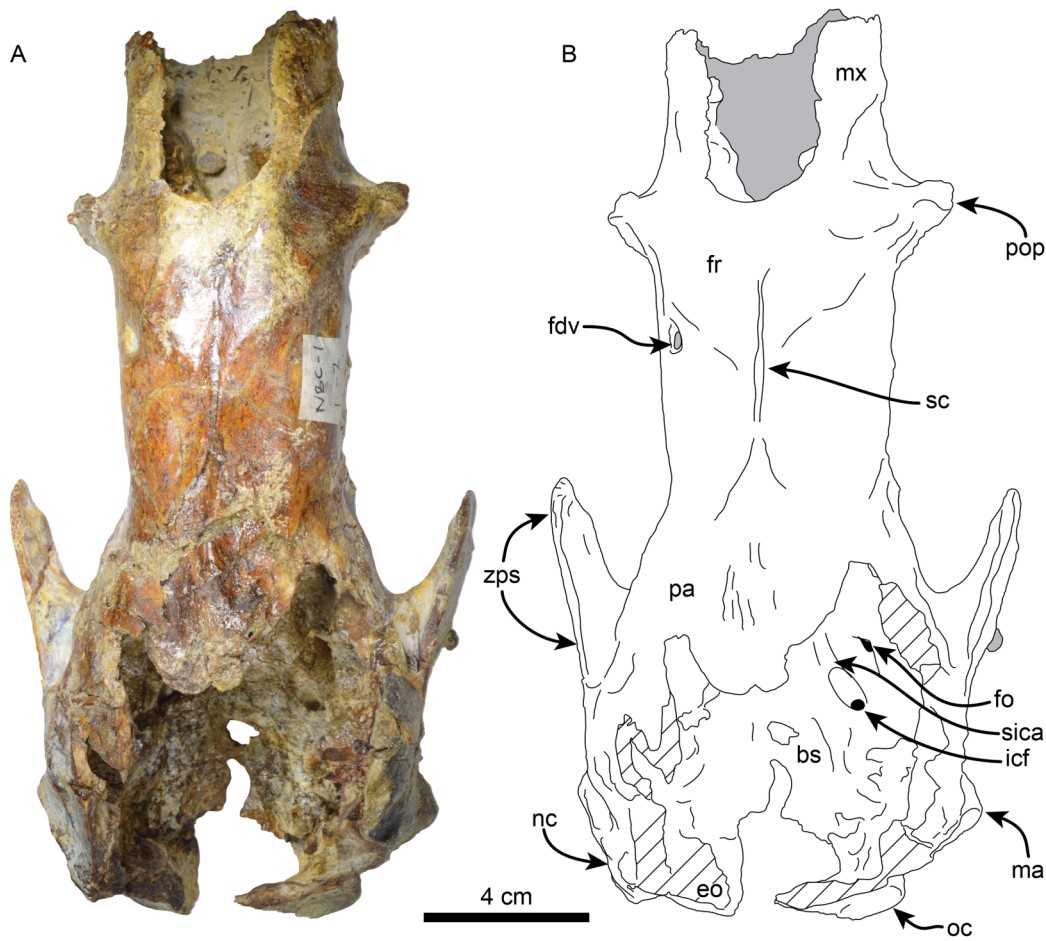

**Figure 2 Skull of _Thalassocnus natans_ (MPC 704-A) in dorsal view (A,B).** Abbreviations: bs, basisphenoid; eo, exoccipital; fdv, foramen for diploic vein; fo, foramen ovale; fr; frontal; icf, internal carotid foramen; ma, mastoid; mx, maxilla; nc, nuchal crest; oc, occipital; pa, parietal; pop, preorbital process; sc, sagittal crest; sica, sulcus for internal carotid artery; zps, zygomatic process of the squamosal. Gray shaded areas are obscured by sediment, diagonal lines denote broken surfaces.

premaxilla, similar to that of _T. antiquus_ (_Muizon et al., 2003_) (Table 2). The lateral surface of the maxilla, anterior to the M1, forms a shallow depression, likely corresponding to the buccinator fossa. The tooth rows are nearly parallel. The palatal portion of the maxilla has a pair of large (~3.5 mm wide by 13 mm long) oval palatal foramina located just medial to M1. Shallow grooves extend anteriorly from the palatal foramina. The infraorbital foramina are situated at the anterior level of M3 and have an oval outline (~10 mm high, 6 mm wide). Their ventral surface is about 5 mm long and approximately 12 mm above the palatal surface. The maxillary foramen is located just posterior to the infraorbital foramen, it has a rounded outline (~4.5 mm in diameter) and opens posteroventrally. On the temporal wall, the suture of the maxilla with the frontal and palatine is interdigitated and slopes posteroventrally. The anterior extension of the palatines is unclear as it is obscured by a conglomerate matrix. There are relatively large (~2.3 mm wide, ~5 mm long), oval,

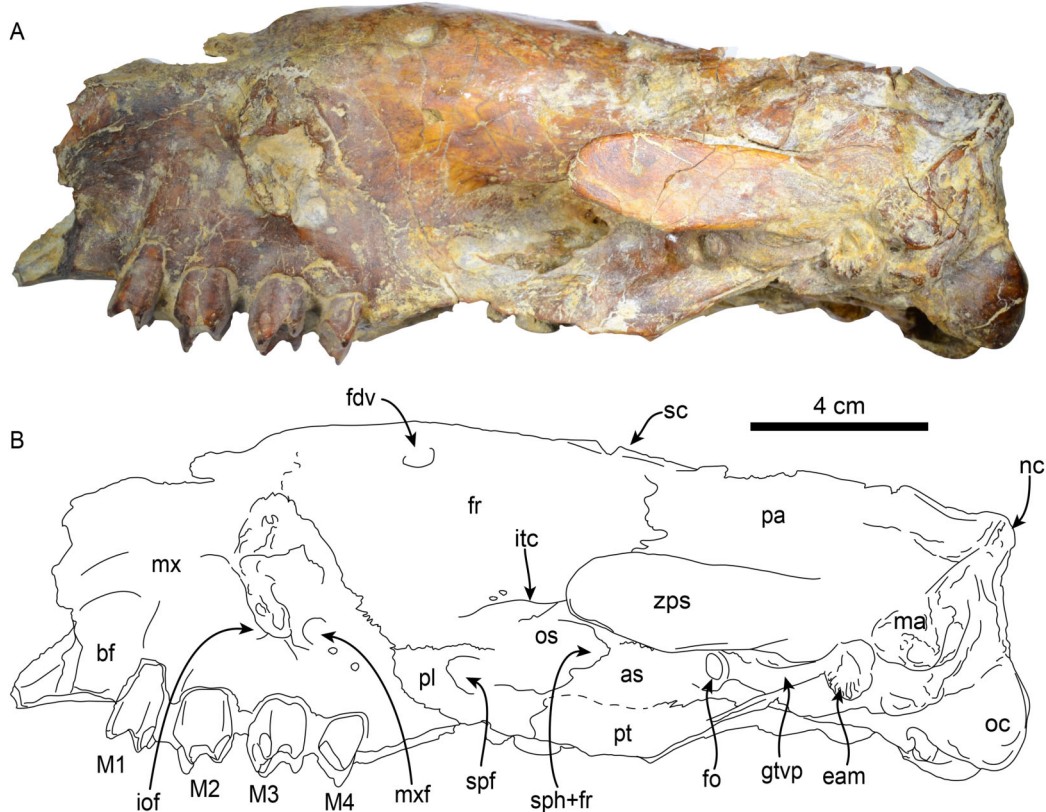

**Figure 3 Skull of *Thalassocnus natans* (MPC 704-A) in left lateral view (A,B).** Abbreviations: as, alisphenoid; bf, buccinator fossa; eam, external auditory meatus; fdv, foramen for diploic vein; fr, frontal; gtvp, groove for m. tensor veli palatini; iof, infraorbital foramen; itc, infratemporal crest; M1–4, first through fourth upper molariforms; ma, mastoid; mx, maxilla; mxf, maxillary foramen; nc, nuchal crest; oc, occipital condyle; os, orbitosphenoid; pl, palatine; pt, pterygoid; spf, sphenopalatine foramen; sph+fr, sphenorbital fissure + foramen rotundum; sc, sagittal crest; zps, zygomatic process of the squamosal.

palatine foramina located approximately at the level of the diastema between M3 and M4. The posterior edge of the palatine is U-shaped in ventral view. In the infratemporal fossa, the palatines contact the maxilla anteriorly, the frontals anterodorsally, the orbitosphenoid posterodorsally and the pterygoid posteriorly. The sphenorbital fissure is rounded and open posterolaterally. Ventrolateral to the sphenorbital fissure, the palatines form part of the long pterygoid processes.

The frontals are elongated and tubular; low temporal lines arise from the contact with the jugal and lacrimal, forming a part of a preorbital process on each side. These lines pass posteriorly and dorsally to the orbital region and converge posterodorsally to join the sagittal crest. A single foramen (~6 mm diameter) for the frontal diploic vein (= supraorbital foramen; *Gaudin, 2011*) is located dorsomedial to the left orbital region (Fig. 3). This foramen seems to be variably present in other xenarthrans, being present only the right side of *Thalassocnus natans* (*McDonald & De Muizon, 2002*: fig. 2) and *Pronothrotherium typicum* (*Gaudin et al., 2020*), but seems to be more consistently

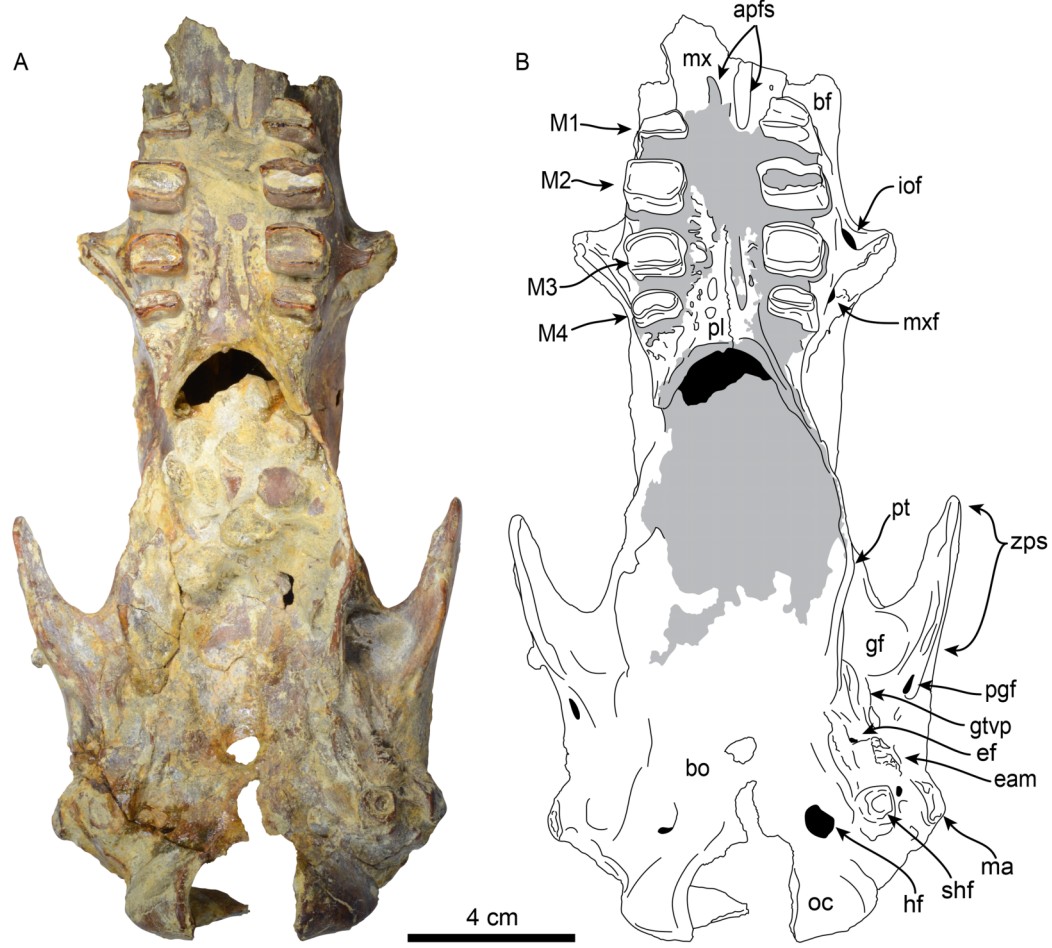

**Figure 4 Skull of *Thalassocnus natans* (MPC 704-A) in ventral view (A,B).** Abbreviations: apfs, anterior palatine foramina; bf, buccinator fossa; bo, basioccipital; eam, external auditory meatus; ef, foramen for Eustachian tube; gf, glenoid fossa; gtvp, groove for m. tensor veli palatini; hf, hypoglossal foramen; iof, infraorbital foramen; M1–4, first through fourth upper molariforms; ma, mastoid; mx, maxilla; mxf, maxillary foramen; oc, occipital condyle; pl, palatine; pt, pterygoid; shf, stylohyal fossa; zps, zygomatic process of the squamosal. Gray shaded areas are obscured by sediment.

bilateral in other taxa (*e.g.*, *Nothrotherium shastense*, *Mionothropus cartellei*, *Neocnus* spp.; *Stock, 1925*; *De Iuliis, Gaudin & Vicars, 2011*; *Gaudin, 2011*). In the temporal wall, the frontals form an anteroposteriorly oriented infratemporal crest that overhangs the orbitosphenoid, alisphenoid and sphenorbital fissures. The frontals contact the maxilla anteroventrally, the orbitosphenoid and alisphenoid ventrally, the squamosal posteroventrally, and the parietals posteriorly and dorsally. The parietals are incompletely preserved. However, a low sagittal crest was present. The contact between the squamosal and frontals is not clearly demarcated, but their profile suggests that they were likely similar to those in *T. natans* (*McDonald & De Muizon, 2002*). The alisphenoid has a large (~4 mm diameter), round foramen ovale. The foramen ovale is bounded dorsally by the alisphenoid-squamosal suture and ventrally by the alisphenoid-pterygoid suture. Posterior

**Table 2  Measurements (in mm) of skull of *Thalassocnus natans* (MPC 704-A).**

| | |
|---|---|
| Basicondylar length from anteroventral angle of ascending process of maxilla to posterior edge of occipital condyles | 217 |
| Length from anterior edge of M1 to posterior of occipital condyles | 205 |
| Length from posterior edge of palate to occipital condyles | 145 |
| Length along midline of palate, from maxilla to palatal incisure | 86 |
| Toothrow length | 54 |
| Length of maxilla anterior to M1 | 25 |
| Maximum width between buccal edges of M1 | 44 |
| Maximum width between buccal edges of M4 | 45 |
| Length from posterior edge of M4 to occipital condyles | 150 |
| Least interorbital width | 57 |
| Maximum width between mastoid processes | 100 |
| Maximum width between lateral margins of occipital condyles | 63 |
| Maximum height of occipital condyles | 29 |
| Width of foramen magnum between ventral edges of occipital condyles | 25 |
| Least distance from posterolingual edge of M4 to post-palatine notch | 8 |
| Maximum width of palate between M1 | 18 |
| Maximum width of palate between M4 | 21 |

**Note:**
Measurements modified from *McDonald & De Muizon (2002)*.

to the foramen ovale, the alisphenoid extends posterodorsally as a thin strip of bone forming the groove for the tensor veli palatini. Anteriorly, the alisphenoid constitutes the posteroventral margin of the sphenorbital fissure and the foramen rotundum. The margins of the orbitosphenoid are not clearly delineated, but its general morphology is similar to that of *T. natans*, forming the temporal wall anterior to the sphenorbital fissure (*McDonald & De Muizon, 2002*). The pterygoid has a subtriangular outline in lateral view, with the transversely thin pterygoid wings flaring ventrolaterally as in *T. natans* (*McDonald & De Muizon, 2002*). Posterodorsally, the pterygoids are not thickened, which differs from the thickened pterygoid wings of the geochronologically younger species *T. littoralis*, *T. carolomartini and T. yaucensis* (*McDonald & De Muizon, 2002*; *Muizon et al., 2004a*). The lateral surface of the pterygoid is shallowly convex. The medial surface is obscured by sediment. No pneumatic sinuses seem to be preserved in the pterygoid surfaces. The zygomatic process of the squamosal is short and anterolaterally oriented; its medial and lateral surfaces are flat with a rounded apex. The glenoid fossa is shallowly concave. A large, oval postglenoid foramen is located posterolateral to the glenoid fossa, near the base of the zygomatic process. The foramen is oriented ventrolaterally as in *Thalassocnus antiquus* and *T. natans*, differing from the ventrally oriented foramen in *T. carolomartini* and *T. yaucensis* (*McDonald & De Muizon, 2002*; *Muizon et al., 2003*, *2004a*; J Velez-Juarbe, 2012, personal observations). The mastoid is rugose and rounded in outline when viewed laterally; its posteroventral margin is medially recurved. Internally, the petrous portion of the squamosal has a large, rounded internal acoustic meatus oriented posterolaterally and located just lateral to the posterior crest of the sella turcica. The

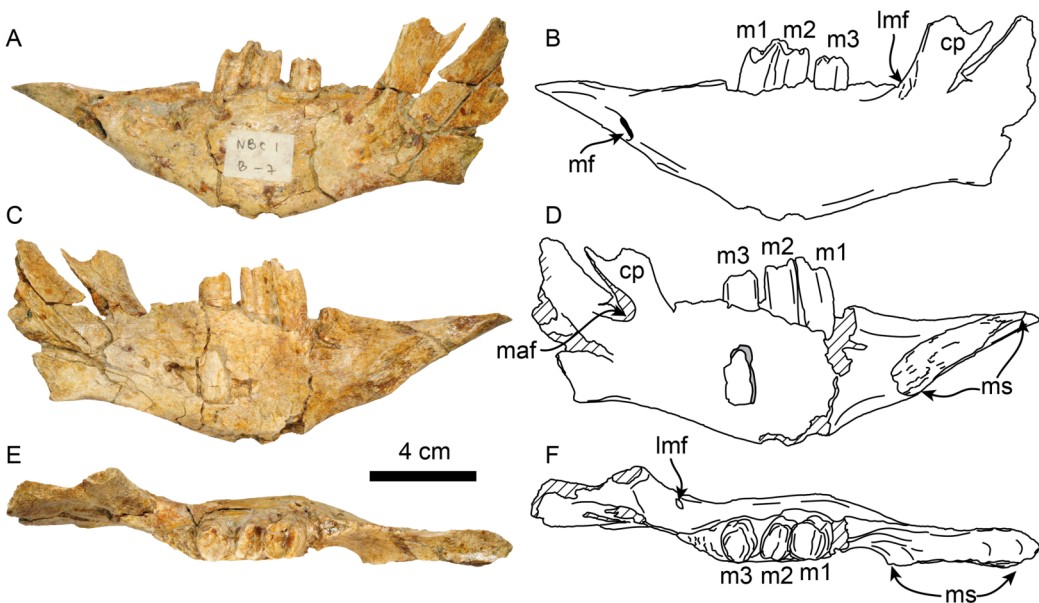

**Figure 5 Dentary of *Thalassocnus natans* (MPC 705) in lateral (A,B), medial (C,D), and dorsal (E,F) views.** Abbreviations: cc, coronoid canal; cp, coronoid process; lmf, lateral mandibular foramen; m1–3, first through third lower molariforms; maf, mandibular foramen; mf, mental foramen; ms, surface of the mandibular symphysis. Diagonal lines denote broken surfaces.

transverse sinus (= crest) arises dorsal to the internal auditory meatus and continues posterodorsally within the medial surface of the bone. The carotid foramen opens internally in an anteromedial direction, and is continuous with a shallow sulcus for the internal carotid artery. Anterolateral to the internal carotid foramen is the similarly-sized internal opening of the foramen ovale. The foramen ovale is large (~4.5 mm in diameter) and is situated ventrolateral to the sphenorbital canal. The ethmoid has a large cribiform plate, suboval in outline, separated by a thick, blunt crista galli. The optic canals in the sphenoid are small, anterolateral, and oval in outline. Posterolateral to these is an enlarged sphenorbital canal, which is suboval in outline. The tuberculum sellae in the basisphenoid is low and convex, with a transverse ridge that has a low clinoid process. The hypophyseal fossa is shallow, the posterior crest is low, with no apparent clinoid processes.

The supraoccipital and exoccipital are incompletely preserved. In lateral view, the nuchal crest is rugose and descends in an anteroventral direction where it joins the mastoid process. The occipital condyles are subtriangular in outline, with their apex at their dorsolateral vertices. The hypoglossal foramen is large (~7 mm in diameter); it is located medial to the fossa for the sternohyoid.

**Dentary**—MPC 705 is an incomplete left dentary, missing a large portion of the coronoid process (Fig. 5; Table 3). The ventral border of the horizontal ramus is ventrally convex below the toothrow, becoming shallowly concave posteriorly. The mandibular foramen opens posterodorsally, it is about 9 mm high by 5 mm wide. The coronoid canal (= lateral mandibular foramen) opens anterolaterally and is located on the lateral surface of the

**Table 3 Measurements of mandible of *Thalassocnus natans* (MPC 705).**

| | |
|---|---|
| Length of mandibular symphysis | 65 |
| Depth of dentary at m2 | 52 |
| Depth of dentary at m3 | 51 |
| Length of the toothrow | 43 |
| Length from the anterior edge of spout to the anterior edge of m1 | 81 |
| Length from the anterior edge of spout to posterior edge of m3 | 124 |
| Width of spout at anterior edge | 28e |

coronoid process at about the same level (height) as the toothrow. There is a shallow fossa on the anteromedial surface of the coronoid process that is interpreted as an insertion site for the m. temporalis. There is a raised triangular, rugose surface just distal to m3. The molars are subrectangular to subrounded in outline, with shallow longitudinal grooves on their buccal and lingual surfaces. The occlusal surfaces of the molariforms are worn, especially along the middle, with their anterior and posterior edges forming transverse crests. The mandibular body tapers dorsoventrally anterior to m1. The symphysis is relatively long (~65 mm in length) and oriented anterodorsally. The anteroventral surface of the symphysis has a relatively straight profile in lateral view resembling the condition observed in *Thalassocnus carolomartini* and differing from the more markedly concave profile observed in *T. antiquus*, *T. natans* and *T. littoralis* (Figs. 5, Fig. S1; *McDonald & De Muizon, 2002*). The mandibular spout is only faintly expanded mediolaterally at about 55 mm anterior from m1, similar to the condition observed in *Thalassocnus antiquus*, *T. natans*, and *T. littoralis*, markedly differing from the transversely expanded spout observed in later species (*Muizon et al., 2004b*; Fig. S1). The predental length of the mandible relative to the toothrow length has a ratio of 1.9, which is proportionally longer than in *T. natans* (1.6; MNHN.F.SAS734) and more similar to that of *T. antiquus* (1.9; MUSM 228). A single and relatively large, oval (9 mm high, 7 mm wide) mental foramen opens anteriorly at about 42 mm anterior to m1.

**Vertebrae**—The axis is nearly completely preserved (Fig. 6). The spinous process is elongated in the craniocaudal direction, and its caudal section is slightly broader than the cranial section, giving this structure a trapezoidal shape in lateral view. The anterior zygapophyseal facet for the articulation with the atlas is subcircular and convex and the postzygapophysis is nearly straight. The vertebral canal is reniform in cranial and caudal views. The vertebral body is dorsoventrally compressed in caudal view. The odontoid process is well-developed, representing approximately a third of the craniocaudal length of the vertebral body, and it is slightly directed cranially and dorsally. Its surface is rugose in the proximal section. The transverse process is poorly developed and forms a short, narrow ridge on the lateral surface of the posterior section of the vertebral body. The lateral opening of the transverse foramen is oval and positioned in the pedicle, posterolateral to the anterior zygapophyseal facet and approximately at the midpoint of the dorsoventral height of the vertebra.

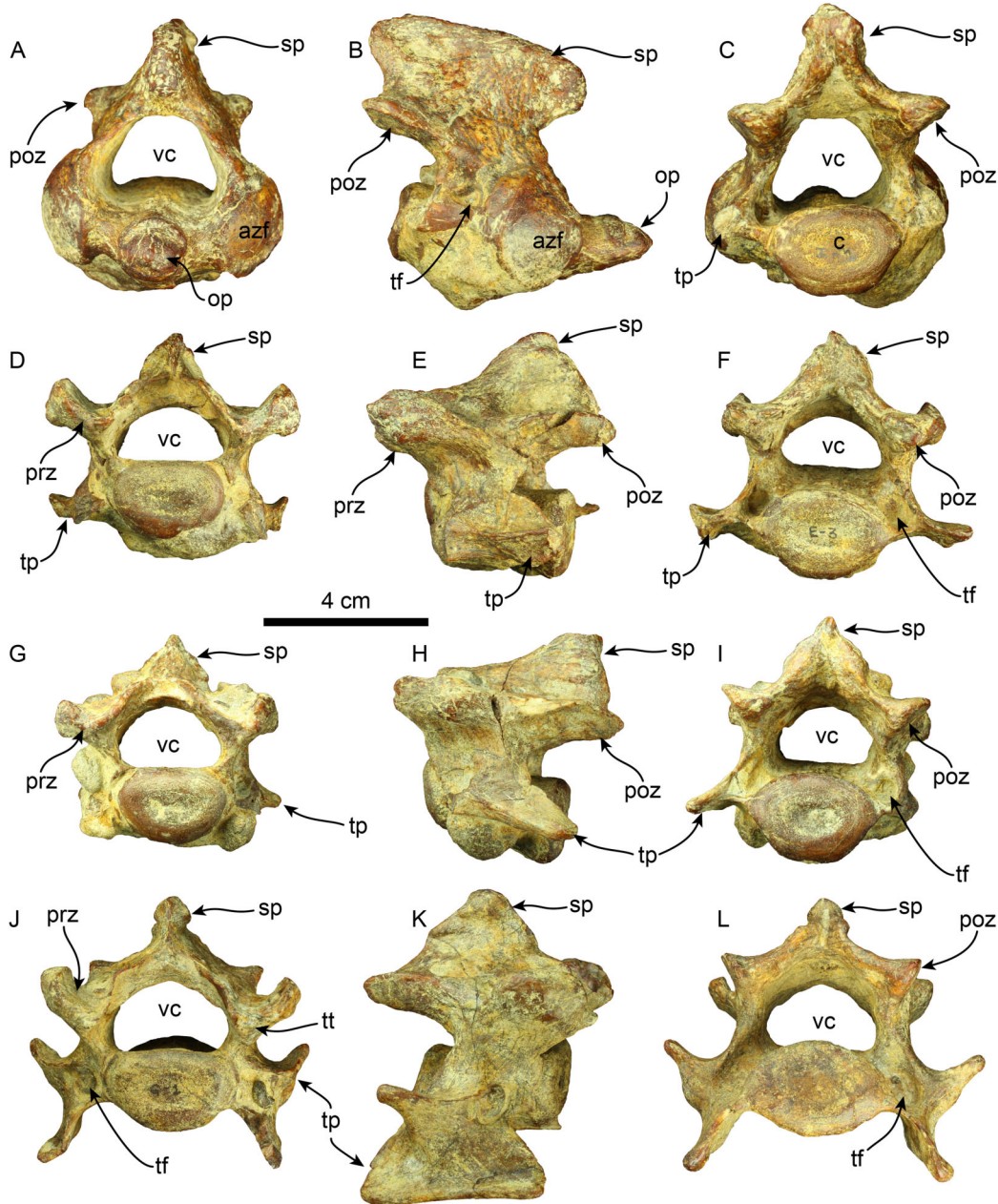

**Figure 6 Cervical vertebrae of *Thalassocnus natans* (MPC 704-A).** Atlas in anterior (A), right lateral (B), and posterior (C), views. Third cervical vertebra in anterior (D), left lateral (E), and posterior (F), views. Fourth cervical vertebra in anterior (G), left lateral (H), and posterior (I), views. Sixth cervical vertebra in anterior (J), right lateral (K), and posterior (L), views. Abbreviations: azf, anterior zygapophyseal facet; c, centrum; op, odontoid process; poz, postzygapophysis; prz, prezygapophysis; sp, spinous process; tf, transverse foramen; tp, transverse process; tt, triangular tuberosity; vc, vertebral canal.

Four other cervical vertebrae are preserved, and are tentatively identified as C3, C4, C6, and C7 (Fig. 6). The spinous processes of C3, C4, and C6 are short, low, and craniodorsally elongated. The spinous process in C7 becomes slightly higher and craniocaudal short,

although this region is partially covered by a hard phosphatic matrix after preparation, obscuring further assessments of this structure. The anterior and posterior zygapophyseal facets are concave and convex, respectively. The cranial side of the C6 prezygapophysis is thickened mediolaterally, forming a tuberosity (*i.e.*, a triangular tuberosity as noted in *Amson et al. (2015c)*). The vertebral canal is reniform, similar to the condition observed in the axis. The vertebral body is dorsoventrally compressed in cranial and caudal views in all cervical vertebrae. The transverse process enlarges in size in caudal direction and progressively shows increased bifurcation along its caudal margin at C6. In contrast, the transverse process in C7 is reduced. The transverse foramen is subcircular, and its size is consistent across the three vertebrae.

A nearly complete set of thoracolumbar vertebrae was found, including 15 thoracic and three lumbar vertebrae (Fig. 7). In the anterior thoracic vertebrae, the spinous process is elongated more dorsoventrally than it is expanded craniocaudally, giving it a rectangular shape. Its height gradually decreases toward the caudal end. The apex of the spinous process is transversally thickened across the thoracolumbar series. The spinous process of the anterior thoracic vertebrae is nearly vertical and gradually increases its caudal inclination (*i.e.*, T12?, T13?) in the more caudal elements until it stabilizes in the posterior thoracic region. There is a well-developed fossa at the base of the spinous process in T17. The anterior zygapophyseal facet is relatively concave in the anterior thoracic vertebrae, contrasting with the flatter state observed in the posterior zygapophyseal facets of both anterior and mid-thoracic vertebrae. The transverse process is positioned dorsally at the mid-level of the pedicle in anterior thoracic vertebrae, transitioning from a more dorsal position to a more ventral one in the mid and posterior vertebrae, respectively. The articular facet for the costal tubercle is located beneath the transverse process in the anterior thoracic region. In the more posterior vertebrae, this facet is flatter on the lateral side of the transverse process. The vertebral canal is relatively circular in the anterior thoracic region but becomes more compressed in the dorsoventral direction in the posterior thoracic and lumbar areas. The vertebral body is also nearly uniformly cylindrical in the cranial and caudal views along the thoracic sequence. The body of the lumbar vertebrae is slightly more dorsoventrally compressed.

Two caudal vertebrae were identified, corresponding to an anterior caudal region located posterior to the sacral region. The spinous process is shorter than that of the preceding vertebrae and extends caudally in lateral view. The prezygapophyses create a flat surface that is dorsolaterally projected. The postzygapophyses are rounded at their terminal section. The transverse processes are nearly straight and located slightly lower than the midpoint of the vertebral body. The vertebral body is cylindrical, with its caudal and cranial surfaces flat. Only the base of the hemapophysis is preserved in one vertebra. The vertebral canal is reniform.

**Radius**—MPC 704-A includes a left radius (Figs. 8A–8H, S2B, S2F; Tables 4, S1). Proximally the head has a rounded outline with a shallowly concave articular surface. The neck is short and subtriangular in cross-section. On the posteromedial surface of the bone, just distal to the neck, there is a well developed bicipital tuberosity, resembling the

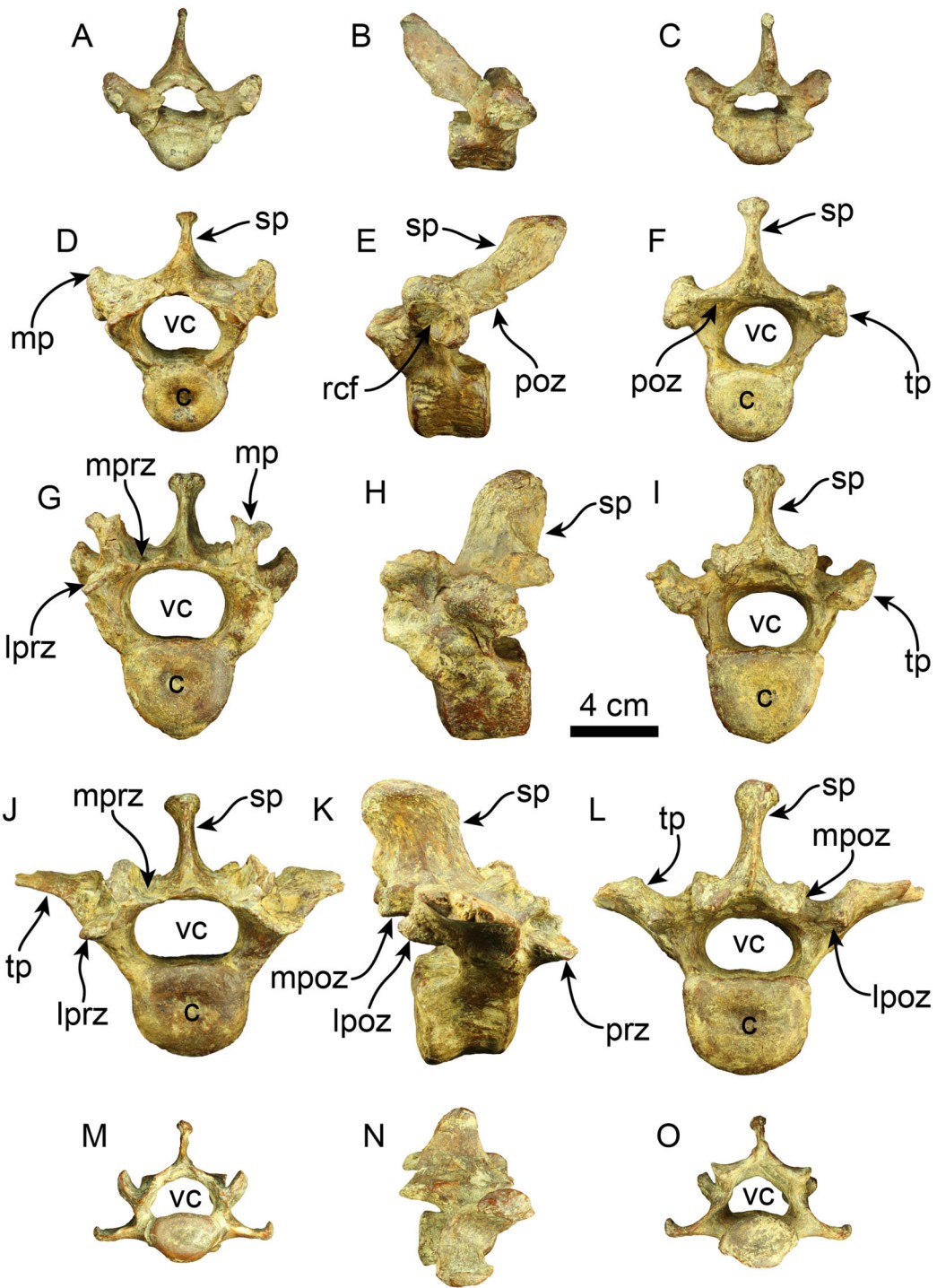

**Figure 7 Thoracic, lumbar and caudal vertebrae of *Thalassocnus natans* (MPC 704-A).** Anterior thoracic vertebra in anterior (A), right lateral (B), and posterior (C), views. Mid thoracic (12th?) vertebra in anterior (D), left lateral (E), and posterior (F), views. Posterior thoracic (17th?) vertebra in anterior (G), left lateral (H), and posterior (I), views. Lumbar (L1?) vertebra in anterior (J), right lateral (K), and posterior (L), views. Anterior caudal vertebra in anterior (M), right lateral (N), and posterior (O), views. Abbreviations: c, centrum; lpoz, lateral postzygapophysis; lprz, lateral prezygapophysis; mp, mammillary process; mpoz, medial postzygapophysis; mprz, medial prezygapophysis; poz, postzygapophysis; rcf, rib capitular facet; sp, spinous process; tp, transverse process; vc, vertebral canal.

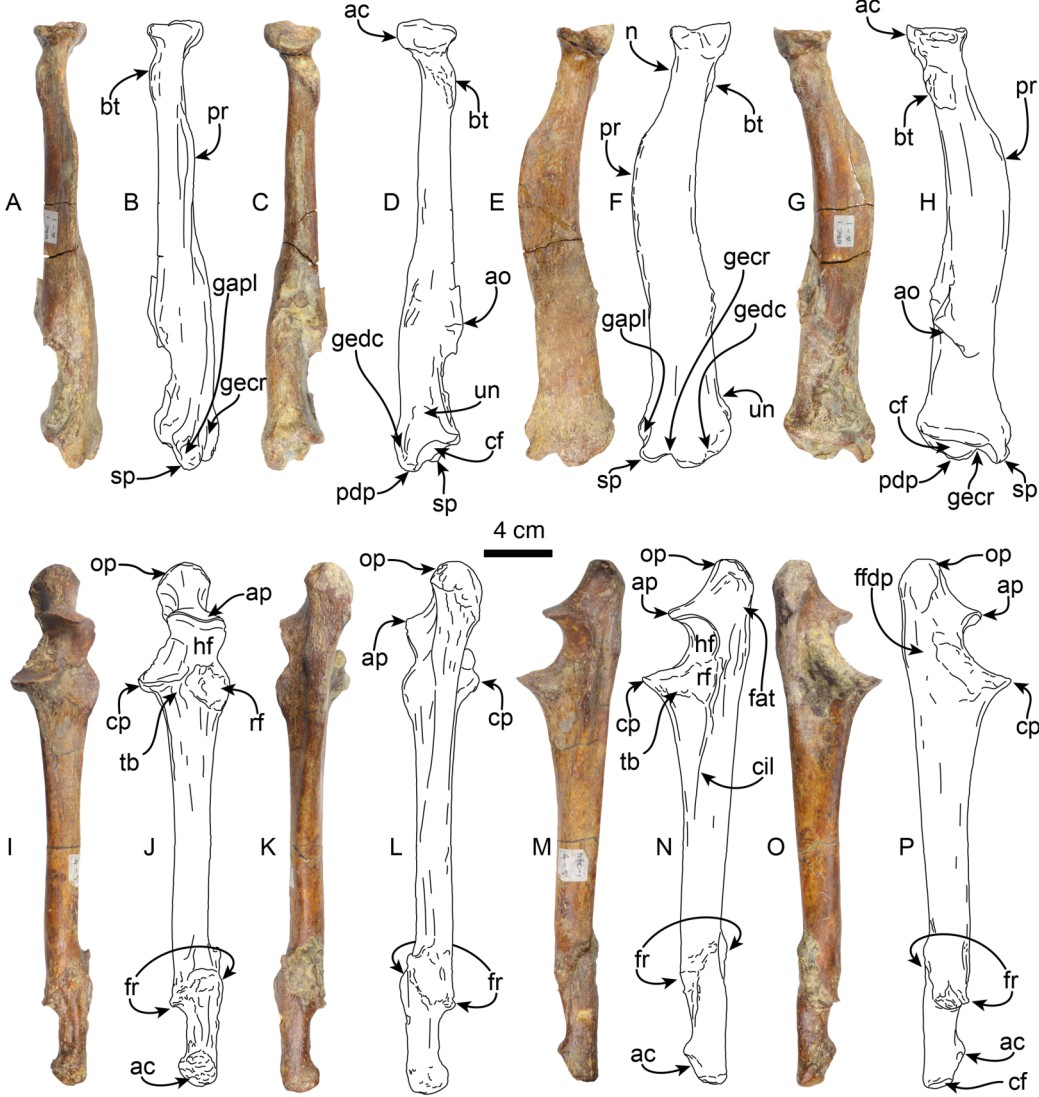

**Figure 8 Radius and ulna of *Thalassocnus natans* (MPC 704-A).** Left radius in anterior (A,B), posterior (C,D), lateral (E,F), and medial (G,H) views. Left ulna in anterior (I,J), posterior (K,L), lateral (M,N), and medial (O,P) views. Abbreviations: ac, articular circumference; ao, accessory ossification; ap, anconeal process; bt, bicipital tuberosity; cf, carpal facet; cp, coronoid process; ffdp, fossa for m. flexor digitorum profundus; fr, healed fracture; gapl, groove for the tendon of m. abductor pollicis longus; gecr, groove for the tendon of m. extensor carpi radialis; gedc, groove for tendon of m. extensor digitorum communis; hf, humeral facet; n, neck; op, olecranon process; pdp, posterodistal process; pr, pronator ridge; rf, radial facet; sp, styloid process; tb, tuberosity for brachialis; un, ulnar notch.

condition observed in *Thalassocnus antiquus* and *T. natans* (*Amson et al., 2015a*). The radial shaft is subtriangular in cross section over most of its length and slightly curved anteriorly, with a large, relatively prominent pronator ridge reminiscent of *Thalassocnus natans* and *T. littoralis*, contrasting with the more attenuated ridge of *T. antiquus* or the more prominent one observed in *T. carolomartini* and *T. yaucensis* (*Amson et al., 2015a*; Fig. S2, Table S1). The shaft of the radius is fractured and healed at about two thirds of the

**Table 4 Measurements (in mm) of the left radius of *Thalassocnus natans* (MPC 704-A).**

| | |
|---|---|
| Maximum length | 264 |
| Length proximal articular surface | 33 |
| Width proximal articular surface | 34 |
| Width distal articular surface | 39 |
| Length distal articular surface | 59 |
| Width at the midshaft | 22 |
| Length at the midshaft | 37 |
| Maximum diameter of capitular fossa | 35 |
| Minimum diameter of capitular fossa | 34 |
| Length of the carpal facet | 58 |
| Width of the carpal facet | 39 |

**Table 5 Measurements in mm of the left ulna of *Thalassocnus natans* (MPC 704-A).**

| | |
|---|---|
| Length, from the olecranon to the styloid process | 301 |
| Length, from the anconeal process to summit of the olecranon | 52 |
| Width at the midshaft | 20 |
| Length at the midshaft | 26 |
| Width of the proximal articular surface | 33 |
| Maximum width of the proximal extremity | 42 |
| Anteroposterior (?) proximal articular surface | 58 |

length of the bone and is slightly offset. As in other species of *Thalassocnus* the anterior and lateral surfaces of the distal end have three prominent tendinal grooves (*Amson et al. 2015a*). The more anterior of these for the tendon of the abductor pollicis longus muscle lies anteromedial to the styloid process (Fig. 8), while the other grooves, for the extensor carpi radialis, and extensor digitorum communis are in a more lateral and posterolateral positions, respectively, and are separated from each other by a prominent posterodistal process (Figs. 8A–8F). The posterodistal process is prominent, extending distally as far as the styloid process as in other species of *Thalassocnus*. The posterior surface at the distal end has a well-defined ulnar notch, that forms a semicircular surface, and thus differing from the more attenuated notch observed in *Thalassocnus natans*, and resembling more the condition observed in later species (*Amson et al., 2015a*). The distal articular surface is partially obscured by sediment, but clearly had an anteroposteriorly oval outline. The groove for the extensor carpi radialis extends into the distal articular surface as in *Thalassocnus natans*, forming a notch on the anterolateral surface of the articulation.

**Ulna**—The left ulna (MPC 704-A) is complete (Figs. 8I–8P, S3A–S3B; Tables 5, S2), and as in the associated left radius there is a broken and healed fracture approximately three-quarters of the way along the shaft; the orientation of the fracture is diagonal to the long axis and the distal quarter of the bone is laterally displaced. The surface along the fracture plane is very rugose and callous. The proximal end of the olecranon is rounded

and rugose, slightly overhanging the medial surface of the bone. The proximal articular surface of the semilunar notch is subrounded in outline, with a low (less than 5 mm) median ridge; the articular surface faces anteriorly. The distal half of the semilunar notch is about half the width of the proximal end and is oriented dorsally. The articular surface for the head of the radius is rounded and lies lateral to the horizontal plane of the semilunar notch. A raised rounded rugose surface lies anteromedially. The shaft is straight and triangular in cross-section proximally, becoming rounded towards the distal end. Distally, the styloid process has a rugose anterior surface. The distal articular surface is rounded and slightly medially oriented. The ulnar proportions of MPC 704-A are similar to those observed in *T. natans* and *T. littoralis*, in contrast to the more robust ulna of *T. yaucensis* (Fig. S3; Table S2).

**Manus**—MPC 704-A includes a partial right manus that includes metacarpal II, fused first and second (ungual) phalanges from the first digit, second and third (ungual) phalanges from the second digit, and fused first and second, and third (ungual) phalanges from the third digit, additionally there are elements of the left hand, including metacarpal III and a third (ungual) phalanx from the second digit (Figs. 9 and 10). Overall, the manus resembles that of *Thalassocnus natans*, *T. carolomartini* (Amson et al., 2015a), and *T. littoralis* (MUSM 223; J Velez-Juarbe, 2012, personal observations). As in other species metacarpal III is longer than metacarpal II and both are close fitting, suggesting a narrow interdigit separation.

The first digit is represented by the fused first and second (ungual) phalanges, and as in other species of *Thalassocnus* it is relatively short and robust (Amson et al., 2015a). The proximal articular surface is shallowly concave in medial view, the subungual process is more prominent than that of the other digits. The ungual process is short, with a triangular profile and not recurved, differing from the more elongated and curved ungual phalanges of digits two and three (Figs. 9 and 10).

The second metacarpal is the smallest of the two preserved metacarpals. The proximal articular surface forms a V-shaped articular surface for the trapezoid, extending from the dorsal to the palmar end of the surface. The proximomedial surface is markedly concave, suggesting a close-fitting contact with the trapezium-first metacarpal complex (MCC), while laterally there is a more prominent semilunar facet for the proximal contact with the adjacent metacarpal III. The shaft is nearly circular in cross section and narrower than the proximal and distal ends. The distal carina is large and smoothly convex in lateral view, and offset towards the lateral half of the distal end, thus further dividing the articular surface into a small lateral shelf and a broader medial shelf. The most dorsal end of the carina extends distally farther than its palmar end. The distal end is offset counterclockwise relative to the proximal end, leaving the palmar border of the carina in a more medial position that its dorsal end. The second phalanx of the second digit has a moderately concave proximal articular surface that is slightly offset counterclockwise, matching the torsion observed in the second metacarpal. The surface is subdivided into two separate surfaces by a midline ridge that extends farther proximally, so that it is visible in lateral or medial views. Distally the shaft becomes dorsopalmarly narrower, then expanding to form

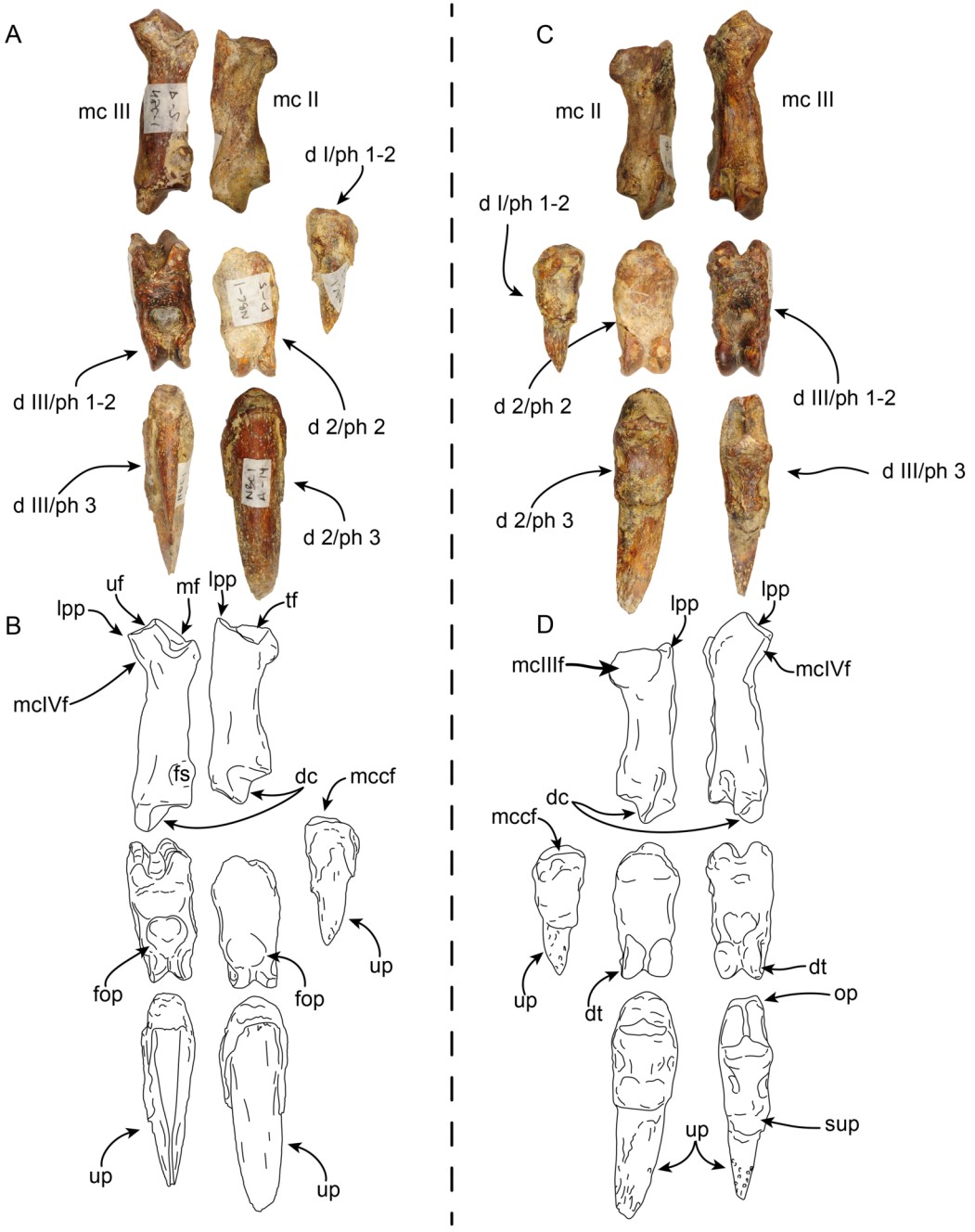

**Figure 9 Partial right manus of *Thalassocnus natans* (MPC 704-A) in dorsal (A,B) and palmar (C,D) views.** Metacarpal III is from the left hand, but reversed for ease of comparison, all other elements are from the right side. Abbreviations: d I/ph 1-2, co-ossified first and second (ungual) phalanges of the first digit; d 2/ph 2, second phalanx of the second digit; d 2/ph 3, third (ungual) phalanx of the second digit; d III/ph1-2, co-ossified first and second phalanges of the third digit; d III/ph 3, third (ungual) phalanx of the third digit; dc, distal carina; dt, distal trochlea; fop, fossa for overhanging process; fs, fused sesamoid; lpp, lateroproximal process; mc II, second metacarpal; mc III, third metacarpal; mcIIIf, metacarpal III facet; mcIVf, metacarpal IV facet; mccf, metacarpal complex facet; mf, magnum facet; op, overhanging process; sup, subungual process; tf, trapezoid facet; uf, unciform facet; up, ungual process.

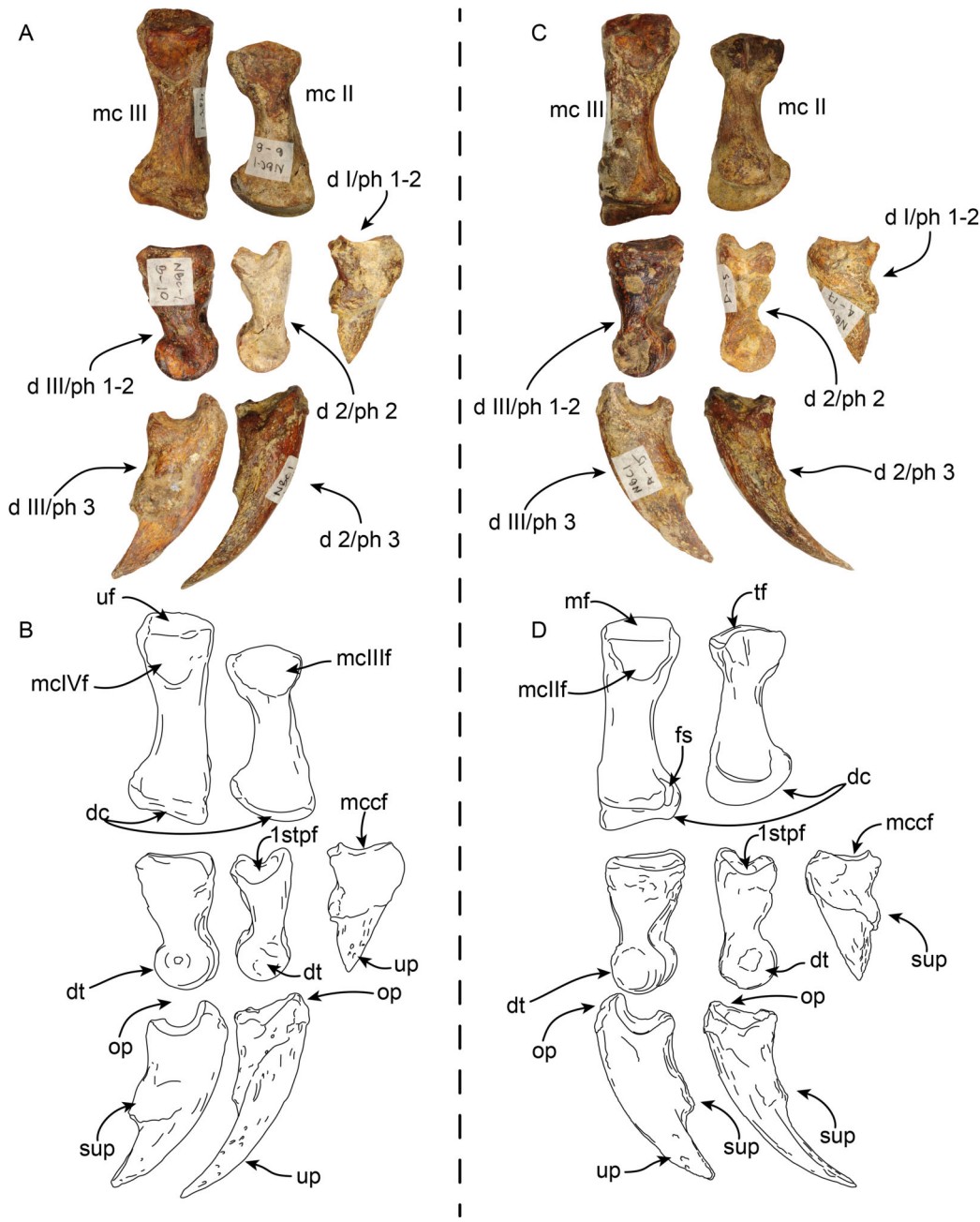

**Figure 10 Partial right manus of *Thalassocnus natans* (MPC 704-A) in lateral (A,B) and medial (C,D) views.** Metacarpal III is from the left hand, but reversed for ease of comparison, all other elements are from the right side. Abbreviations: d I/ph 1-2, co-ossified first and second (ungual) phalanges of the first digit; d 2/ph 2, second phalanx of the second digit; d 2/ph 3, third (ungual) phalanx of the second digit; d III/ph1-2, co-ossified first and second phalanges of the third digit; d III/ph 3, third (ungual) phalanx of the third digit; dc, distal carina; dt, distal trochlea; fs, fused sesamoid; mc II, second metacarpal; mc III, third metacarpal; mcIIf, metacarpal II facet; mcIIIf, metacarpal III facet; mcIVf, metacarpal IV facet; mccf, metacarpal complex facet; mf, magnum facet; op, overhanging process; sup, subungual process; tf, trapezoid facet; uf, unciform facet; up, ungual process.

the distal trochlea. The dorsal surface of the shaft has a relatively deep, rounded fossa to accommodate the overhanging process of the ungual phalanx, whereas ventrally there is a relatively shallower fossa. Altogether these form a nearly circular trochlea (~270°) in lateral view; in dorsal or ventral views the trochlea is transversely narrower than the shaft, resembling the condition observed in other species of *Thalassocnus* (*Amson et al., 2015a*). The ungual phalanx of the second digit is elongated (longest in the series) with an ungual process that is long and relatively flat (dorsopalmarly), and gently curved medially and ventrally. As in the preceding phalanx, the proximal articular surface forms a relatively deep concavity that is subdivided by a median ridge that extends proximally farther than the lateral and medial edges of the articular surface.

Metacarpal III (Figs. 9 and 10) is elongated and nearly straight with similarly sized proximal and distal ends. The proximal articular surface is characterized by multiple articular facets and a prominent lateroproximal process. The lateral surface has a flat, nearly triangular facet for metacarpal IV that is separated by a low ridge from the rectangular facet for the unciform. Medially, there is a subrounded, concave facet for metacarpal II that is divided from the facet for the magnum by a low ridge. The shaft becomes narrower when viewed medially or laterally, becoming expanded towards the distal end. The distal end is characterized by a distal carina that is shifted laterally that, as in metacarpal II, divides the articular surface into a small lateral shelf and a broader medial shelf. The distal carina is large, but with a vertical profile in lateral view, differing from the broadly arched profile of the carina in metacarpal II. In distal view the carina is curved medially. The first and second phalanx of the third digit are co-ossified resembling the condition observed in other specimens of *Thalassocnus* (*Amson et al., 2015a*). In palmar view, a faint transverse line marks the joint between these two elements. The proximal articular surface is broadly V-shaped in dorsal or palmar view, given the more vertical profile of the distal carina of metacarpal III there would have been limited vertical movement of this element. The dorsal and palmar surfaces have a relatively deep fossa for the overhanging process of the ungual phalanx. The distal articular surface is expanded, forming a nearly circular trochlea. In dorsal or palmar views, the trochlea is transversely narrower than the rest of the bone. The ungual phalanx of the third digit is stouter, but shorter than that of the second digit. Proximally it has a deeper articular surface and a more prominent overhanging process. The ungual process is shorter, but with a similar curvature as that of digit two.

**Pelvis**—MPC 704-A comprises an incomplete left pelvis, retaining part of the iliac wing, the acetabulum, and a fragment of the ischium (Fig. 11). The preserved iliac wing is flat in laterodorsal view, with a hard phosphatic matrix partially obscuring its dorsal surface. The acetabulum exhibits a suboval shape and a distinct ventral rim along its edge. There is a minor tuberosity in the lateral surface, located ~1 cm anterior to the anterior margin of the acetabulum, likely representing the attaching area of the m. rectus femoris. The medial border of the sacro-ischiatic fossa is preserved and is located at the mid-level of the acetabulum.

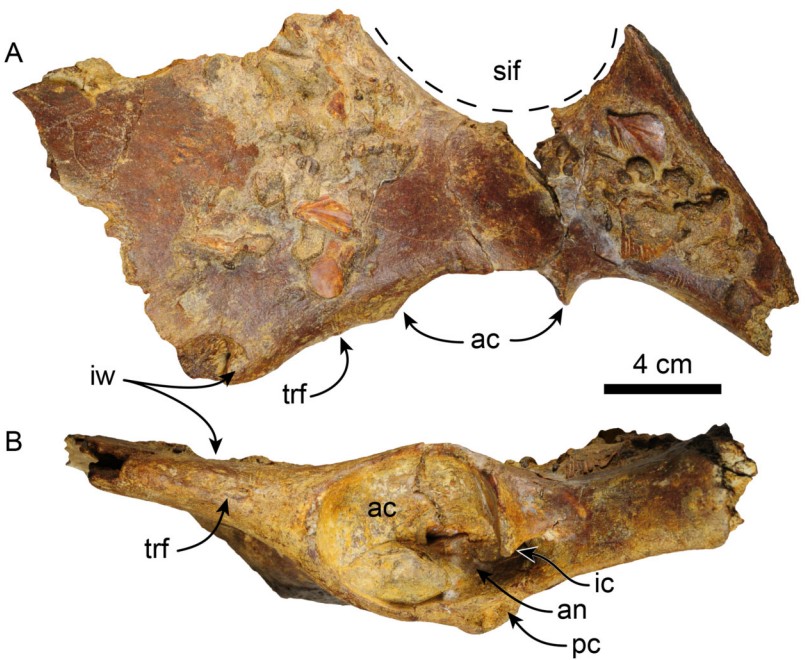

**Figure 11 Left pelvis of *Thalassocnus natans* (MPC 704-A) in dorsal (A) and lateral (B) views.**
Abbreviations: ac, acetabulum; an, acetabular notch; ic, ischiatic cornu; ipe, iliopectineal eminence; iw, iliac wing; pc, pubic cornu; sif, sacro-ischiatic fossa.

**Femur**—The right femur of MPC 704-A is complete (Figs. 12, S4A, S4F; Tables 6, S3–S5). The shaft is anteroposteriorly flattened, transversely narrow and curved posteriorly. The head is oriented dorsomedial to the long axis of the shaft, and the neck is short. The head is rounded, with the fovea capitis forming a triangular indentation on the posteromedial surface. The greater trochanter does not rise above the level of the head. Its dorsolateral surface is wide and rugose; the trochanteric fossa is elongated (39 mm high for 12 mm wide) and deep (~45 mm). A prominent parasagittal ridge (~37 mm long) descends distally from the anterior surface of the greater trochanter, reaching about the level of the lesser trochanter, which likely limits the insertion area of the m. vastus lateralis (*Amson et al., 2015b*: fig. 5). The third trochanter is continuous with the lateral surface of the greater trochanter; its surface is rugose and extends laterally farther than the greater trochanter; it overhangs the anterior surface of the shaft. The lesser trochanter begins at about 45 mm distal to the head. Its surface is rugose and extends along the medial surface of the bone for about 69 mm. The distal end is wider than the proximal end, as is characteristic of *Thalassocnus* (*Amson et al., 2014*). The patellar surface is continuous with the condyles, and its medial edge is more prominent. The medial and lateral epicondylar ridges are present; the medial ridge continues over the anteromedial surface, serving as the insertion site for m. adductor magnus; the ridge gradually slopes towards the epicondyle. The lateral ridge is less prominent. The medial epicondyle is rugose and rounded in outline; the lateral epicondyle is less rugose. The lateral condyle is subrectangular in outline with a mediolateral flat surface. The medial condyle is oval in outline, rounder than the lateral condyle, and extending farther distally and posteriorly. The morphology of MPC

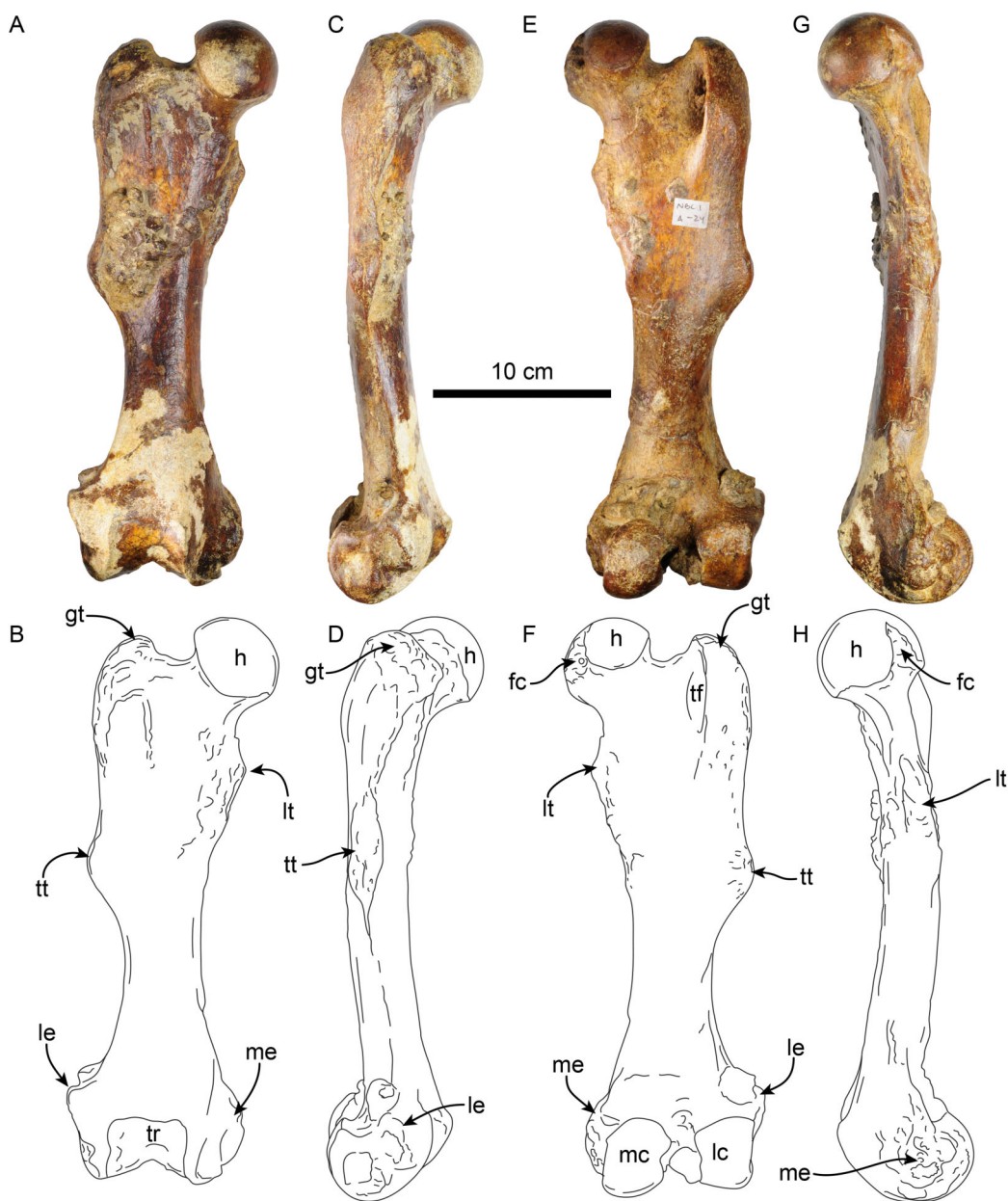

**Figure 12 Right femur of *Thalassocnus natans* (MPC 704-A) in anterior (A,B), lateral (C,D), posterior (E,F), and medial (G,H) views.** Abbreviations: fc, fovea capitis; gt, greater trochanter; h, head; lc, lateral condyle; le, lateral epicondyle; lt, lesser trochanter; mc, medial condyle; me, medial epicondyle; tf, trochanteric fossa; tr, trochlea; tt, third trochanter.

704-A is nearly identical to that of MPC 644 from Cerro Ballena (*Pyenson et al., 2014*; Fig. S4; Table S3). Both are relatively slender with length to midshaft width ratios closer to those measured for *T. littoralis* (Table S3).

    The other right femur (MPC 705-A; Figs. S4B, S4G; Tables 6, S3–S5) is slightly shorter, more robust and has an anteroposteriorly wider third trochanter than that of MPC 704-A.

**Table 6 Measurements (in mm) of femora of *Thalassocnus natans* (MPC 704-A and MPC 705-A).**

|  | MPC 704-A | MPC 705-A |
|---|---|---|
| Length, maximum | 313 | – |
| Maximum distance from the greater trochanter to the lesser trochanter |  | – |
| Maximum proximal width |  | – |
| Width of the patellar groove | 44 | – |
| Maximum width of distal articular surface | 82 | 92 |
| Distance between condyles | 22 | 19 |
| Width at mid shaft | 47 | 48 |
| Length at mid shaft | 32 | 32 |
| Maximum width between ento and ectopicondyles | 102 | – |
| Anteroposterior femoral head diameter | 55 | – |

The lateral supraepicondylar ridge also differs from that of MPC 704-A, being more prominent and flange-like. The medial condyle is wider, flatter and elongated in outline, whereas the lateral condyle is more rectangular. Both condyles show less size and morphological disparity than those of MPC 704-A.

**Tibia**—The shaft of the left tibia (MPC 704-A) is circular in cross-section and slightly bowed medially (Fig. 13). The proximal and distal epiphysis are wider transversely than anteroposteriorly deep. As with the femur, the tibia is relatively slender, but stands out by its length, which approximates that of the femur (Tables 6, 7). The proximal articular surface is separated by a groove along the midline into medial and lateral facets. The lateral facet is subtriangular in outline with a nearly flat to shallowly convex surface; while the lateral facet has a prominent posterolateral border, it however does not seem to correspond to the fused cyamo-fabella observed in other species of *Thalassocnus* (*Salas, Pujos & de Muizon, 2005*; *Amson et al., 2014*). Distolateral to the lateral condylar facets, a prominent oval, flat surface marks the proximal fibular articulation. The medial articular surface is round and deeply concave. Proximally on the anterior surface of the tibia is a prominent, transversely wide tibial tuberosity for insertion of m. quadriceps femoris (*Toledo, Bargo & Vizcaíno, 2015*). A low tuberosity is observed on the medial surface of the shaft, just proximal to the midlength of the bone, likely the attachment site of m. biceps femoris (*Toledo, Bargo & Vizcaíno, 2015*). Distally, the articular surface for the astragalus is divided by a prominent anteroposterior median ridge into the odontoid facet and the larger, discoid facet, resembling the condition observed in most species of *Thalassocnus* (*Amson et al., 2014*). The odontoid facet is relatively high and open anterodistally. The discoid facet is broadly semilunar in outline, shallower, but larger than the odontoid facet. The fibular facet is anteroposteriorly elongated and faces posterolaterally. Along the medial surface, just proximal to the facet for the odontoid process, there are three distinct ridges from the grooves for the digital long flexors (m. tibialis caudalis, flexor digitorum longus and flexor hallucis longus). As in *Thalassocnus natans* the groove for the m. flexor digitorum

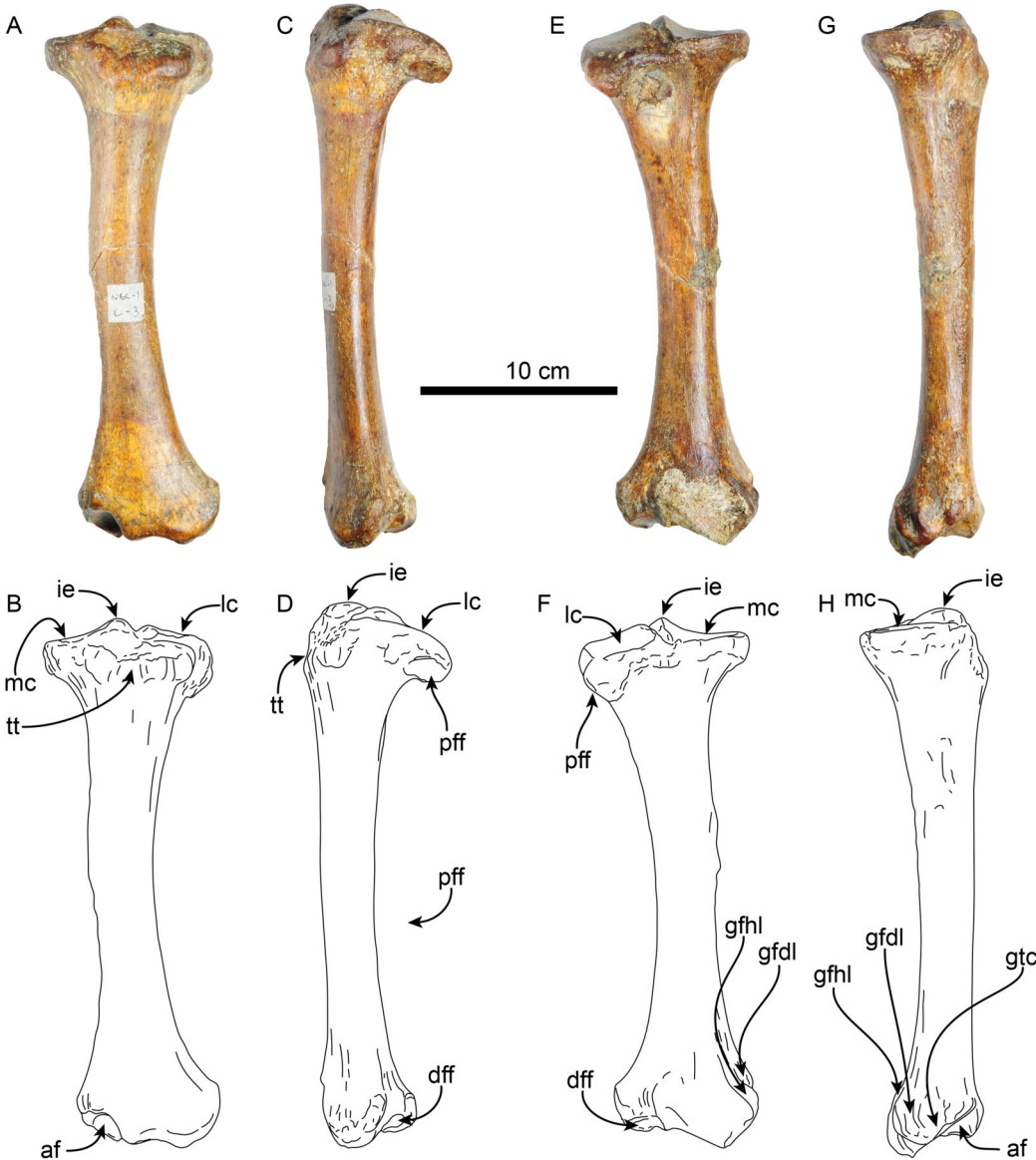

**Figure 13 Left tibia of *Thalassocnus natans* (MPC 704-A) in anterior (A,B), lateral (C,D), posterior (E,F), and medial (G,H) views.** Abbreviations: af, astragalus facet; dff, distal facet for the fibula; gfdl, groove for tendon of m. flexor digitorum longus; gfhl, groove for tendon of m. flexor hallucis longus; gtc, groove for tendon of m. tibialis caudalis; ie, intercondylar eminence; lc, lateral condyle; mc, medial condyle; pff, proximal facet for the fibula; tt, tibial tuberosity.

longus extends along a shallow tubercle and is bounded by more prominent ridges (Figs. 13G–13H).

A second left tibia (MPC 705-A; Table 7) from the Norte Bahía Caldera is generally similar to MPC 704-A. However, as with the femur, it is more robust and slightly larger in its overall dimensions. At the proximal end the medial articular surface is wider, matching

| Table 7 Measurements (in mm) of tibiae of *Thalassocnus natans* (MPC 704-A and MPC 705-A). | | |
| --- | --- | --- |
| | **MPC 704-A** | **MPC 705-A** |
| Maximum length | 267 | 282 |
| Width proximal articular surface | 85 | 97 |
| Length proximal articular surface | 56 | – |
| Width distal articular surface | 73 | 80 |
| Length distal articular surface | 46 | 54 |
| Width at midshaft | 33 | 34 |
| Length at midshaft | 27 | 30 |

the more prominent medial condyle of the associated femur (MPC 705-A). The distal fibular facet is more posteriorly oriented and more prominent, while the facet for the odontoid appears flatter.

**Pes**—MPC 704-A includes several elements of the right pes (astragalus, calcaneus, metatarsal V a co-ossified phalanges 1-2 of the second digit (Figs. 14, 15). The right astragalus has a prominent, odontoid process, with a cylindrical outline, and oriented perpendicular to the discoidal facet (Fig. 14) as in *Thalassocnus antiquus* and *T. natans* (*Amson et al., 2014*). The discoidal facet is broadly expanded laterally as in most species of *Thalassocnus* (*Amson et al., 2014*). A roughly triangular fibular facet is on the lateral surface of the astragalus at about nearly 90° from the discoidal facet; a low ridge separates these otherwise nearly continuous surfaces. The navicular process is short, with a rounded navicular facet. The navicular facet can be divided into two parts, a proximolateral deeply concave fossa, and a semilunar plantar portion that forms a smooth shallowly convex surface (Figs. 14A–14H). The ventral edge of the navicular facet is nearly continuous, with the cuboid facet on the plantar surface of the bone. The sustentacular facet is small, and nearly continous with the cuboid facet, both are separated from the ectal facet by a relatively broad sulcus tali (Figs. 14C and 14D). The ectal facet is shallowly concave, with an oval outline and well-defined edges, laterally it approximates the ventral edge of the fibular facet.

The calcaneus is strongly curved in lateral view (Figs. 14I and 14J). The tuber calcis is broad, with a subtriangular outline in dorsal and lateral views. The medioproximal process lies in a more anterior position relative to the lateroproximal process. The dorsal surface of the tuber calcis has a tuberosity, interpreted as the insertion for the m. gastrocnemius. The neck of the calcaneus is short and narrow with a circular cross section. The distal end of the calcaneus has three, well-defined articular facets. The more dorsal and largest of these, the ectal facet, is oriented obliquely (~45° to the horizontal plane); its surface is convex and oval in outline; it is separated from the sustentacular facet by a transversely sinuous (in distal view) sulcus calcanei (Figs. 14O and 14P). The sustentacular facet of the astragalus is oriented dorsomedially and separated from the smaller cuboid facet by a transverse ridge. The cuboid facet is anteriorly and mediolaterally oriented, elongated and concave. On the

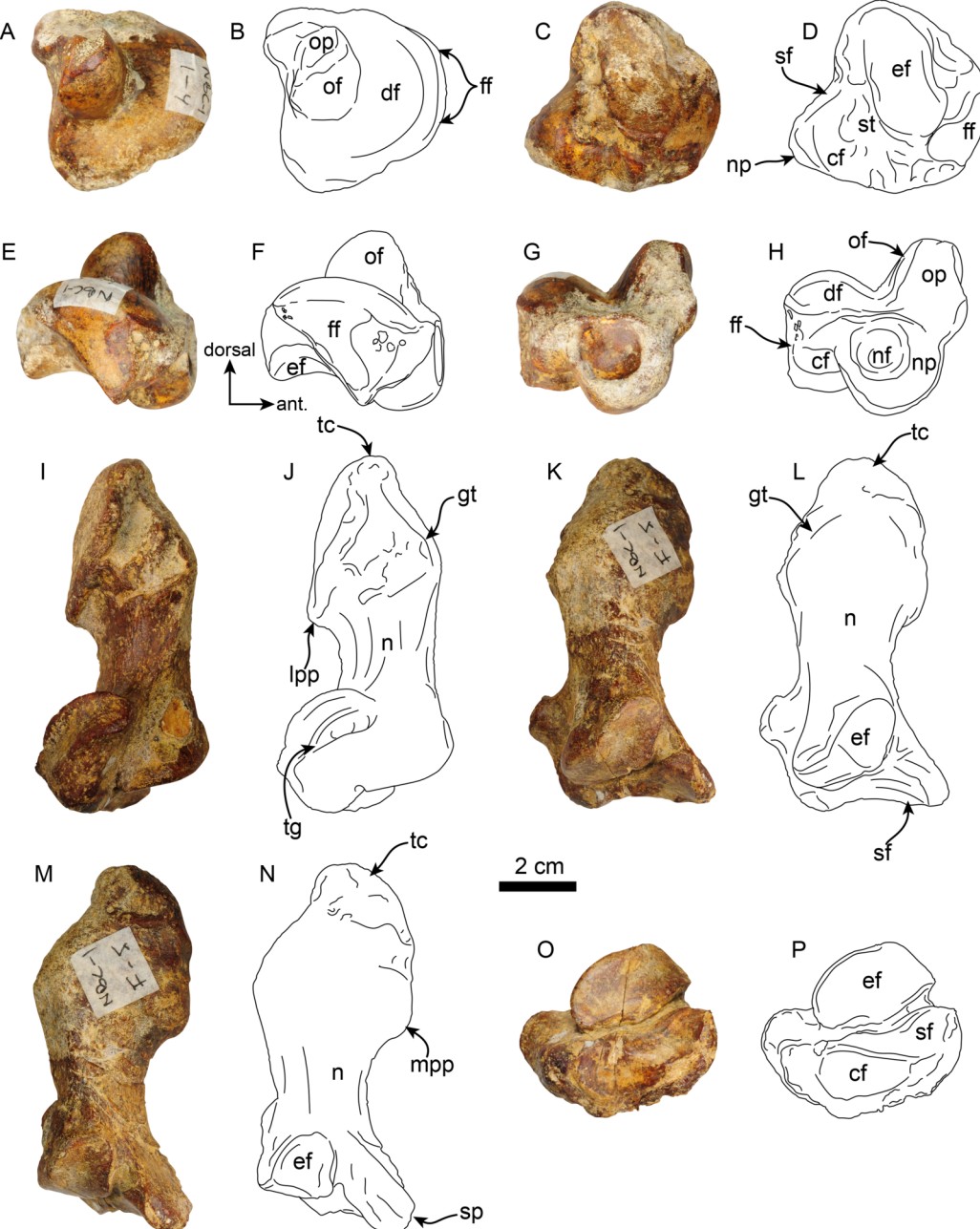

**Figure 14 Astragalus and calcaneus of *Thalassocnus natans* (MPC 704-A).** Right astragalus in dorsal (A,B), plantar (C,D), fibular (E,F), and distal (G,H) views. Righ calcaneum in lateral (I,J), dorsal (K,L), medial (M,N), and distal (O,P) views. Abbreviations cf, cuboid facet; df, discoid facet; ef, ectal facet; ff, fibular facet; gt, gastrocnemial tuberosity; lpp, lateroproximal process; mpp, medioproximal process; n, neck; nf, navicular facet; np, navicular process; of, odontoid facet; op, odontoid process; sf, sustentacular facet; sp, sustentacular process; st, sulcus tali; tc, tuber calcis; tg, tendinal groove.

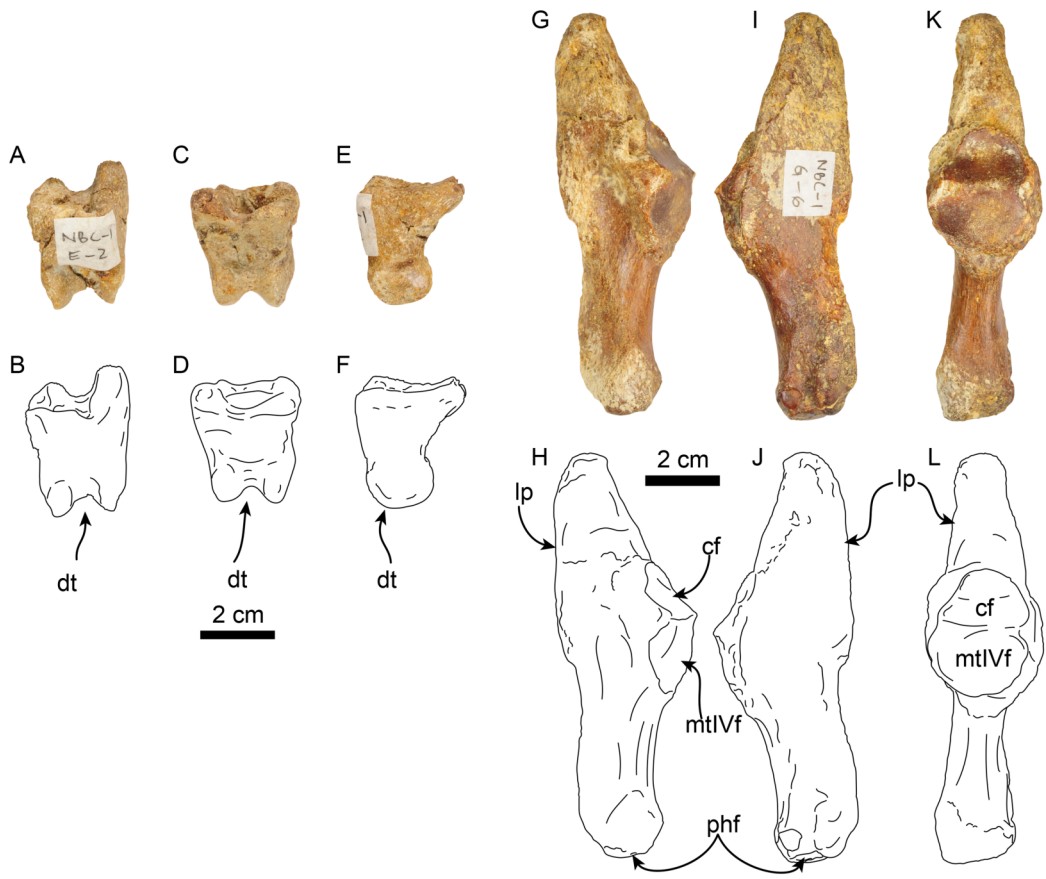

**Figure 15 Phalanx and metatarsal V of *Thalassocnus natans* (MPC 704-A).** Right co-ossified 1st and 2nd phalanges of the second right digit in dorsal (A,B), plantar (C,D), and medial (E,F) views. Right metatarsal V in dorsal (G,H), plantar (I,J), and medial (K,L) views. Abbreviations: cf, cuboid facet; dt, distal trochlea; lp, lateral process; mtIVf, metatarsal IV facet; phf, phalangeal facet.

lateral surface near the distal end, there is a robust, prominent process that has a deep, oblique tendinal groove (Figs. 14I and 14J), resembling the condition observed in *Thalassocnus natans* (*Amson et al., 2014*).

The co-ossified 1st and 2nd phalanges of the second pedal digit are preserved (Figs. 15A–15F). This bone has not been previously described for *T. natans*, but its overall morphology is similar to those of *T. littoralis* and *T. carolomartini* (*Amson et al., 2014*). The proximal articular surface is uneven, with a longitudinal dorsoplantar groove that would have accomodated the distal carina of metatarsal II. The distal end has a poorly developed distal trochlea and the ventral surface has a very shallow fossa, all suggestive of limited movement along the axis of the trochlea.

The right metatarsal V is typical of *Thalassocnus*, particularly with *T. natans*, given the level of development of the lateral process (*Amson et al., 2014*). As in *T. natans* the lateral process is well expanded proximally, and the lateral edge of the bone is nearly rectilinear (Figs. 15G–15J). The articular facets for the cuboid and metatarsal IV are oriented medially and slightly dorsally, as in other species of *Thalassocnus* (*Amson et al., 2014*); the articular

surfaces are separated from each other by a transverse ridge and form an obtuse angle. The articular surfaces are nearly flat and circular, the cuboid facet is slightly smaller. Distal to the articular surface the shaft has an oval outline, being flatter dorsoventrally than mediolaterally. The distal end is more expanded and knob-like, the articular surface for the proximal phalanx is relatively flat.

## DISCUSSION

### Taxonomic identity

The material from Norte Bahía Caldera resembles two species of *Thalassocnus*, *T. natans* and *T. littoralis*. Based on our observations we provisionally assign the material described here (MPC 704-A, MPC 705, and MPC 705-A) to *Thalassocnus natans*, ruling out earlier and later species. The Norte Bahía Caldera skull and mandible has more similar dimensions with *T. natans* than with the smaller *T. antiquus* or the larger *T. littoralis*. The mandible (MPC 705) has a relatively linear anteroventral border of the symphysis, which differs from the more concave border of *Thalassocnus antiquus*, *T. natans*, and *T. littoralis*, but resembling *T. carolomartini*. The radius (MPC 704-A) has an enlarged bicipital tuberosity that is similar to *T. antiquus* and *T. natans*, differing from the smaller tuberosity in later species. Additionally, a relatively prominent pronator ridge is observed in MPC 704-A, thus differing from the more discrete ridge in *T. antiquus* and the prominent crest-like ridge of later taxa (*T. carolomartini* and *T. yaucensis*; *Muizon et al., 2003*; *Amson et al., 2015a*), being instead closer to the condition observed in *T. natans* and *T. littoralis* (Fig. S2; Table S1). The apex of the spinous process of the thoracic vertebrae is transversely thickened across the thoracolumbar series, similar to *T. natans* and differing from *T. littoralis*, *T. carolomartini*, and *T. yaucensis*. Femoral dimensions for MPC 704-A and MPC 644 suggest a proportionally slender femur, with length to width ratios similar to those observed in *T. littoralis* (Table S3). Similarly, the ratio of femur length to skull size is more akin to those of *T. littoralis* than to other taxa sampled (Table S4), but the femoral length to tibial length ratios is much closer to that of *T. natans* (Table S5). Other aspects of the postcranial skeleton, such as the morphology of the tibia and metatarsal V are consistent with *T. natans*. Meanwhile, differences between the more slender femora of MPC 704-A and MPC 644 with MPC 705-A could be indicative of intraspecific variation or sexual dimorphism. Previous workers have already suggested that *Thalassocnus* is sexually dimorphic (*Muizon et al., 2004a*; *Amson et al., 2014*, *2015b*), and sexual dimorphism has also been documented for other extinct folivorans (*e.g.*, *McDonald, 2006*; *Boscaini et al., 2019*; *Cartelle et al., 2019*; *McAfee & Beery, 2021*). Nevertheless, while the unique mosaic of morphological features observed in the Norte Bahía Caldera specimens could be considered to represent a new taxon, additional associated material is still needed to more thoroughly assess the degree of intraspecific or regional variation in *Thalassocnus*.

### *Thalassocnus* records from the eastern coast of the South Pacific

*Pisco Formation*–The original descriptions of all five species belonging to the genus *Thalassocnus* refer to late Neogene material collected from marine sediments of the Pisco Formation in Peru (Table 8). This formation spans the East Pisco Basin (EPB) in the Ica

Valley and is part of a series of coastal sedimentary sequences deposited across shoreface and offshore shelf environments. These sequences have produced abundant, well-preserved, and diverse fossil marine vertebrates, including cetaceans, pinnipeds, seabirds, sharks, and fishes (*Ochoa et al., 2021*, *2022*). In the EPB, the Pisco Formation unconformably overlies basement rocks, the Eocene-Oligocene age Paracas and Otuma formations, or the Miocene age Chilcatay Formation, and it is capped by Plio-Pleistocene marine sediments. Further south from the Ica Valley between the towns of Puerto Lomas and Yauca, the southern part of the East Pisco Basin crops out in the Sacaco sub-basin (or area), where the Pisco Formation is capped by the Pliocene age Caracoles or Pongo formations. The fossil-bearing localities in the Sacaco area are dispersed widely from one another, covered by aeolian sands, and vary widely in stratigraphic thickness across the area. These challenges have made basin-wide age correlations in the EPB difficult across fossil-bearing localities (see *Ochoa et al., 2021*, and references therein). To date, *Thalassocnus* material from the Pisco Formation of Peru has only been reported from the Sacaco area.

Various studies (*e.g.*, *Muizon & DeVries, 1985*; *Ehret et al., 2012*; *Lambert & De Muizon, 2013*) have described five main fossil-bearing horizons in the Sacaco area: (1) El Jahuay (ELJ), Aguada de Lomas (AGL), Montemar (MTM), Sud-Sacaco (SAS), and Sacaco (SAO). *Thalassocnus antiquus* is the oldest species of the genus, and its type specimen (MUSM 228) was collected from the Aguada de Lomas (AGL) locality from marine strata with a vertebrate fossil-bearing horizon of the same name (AGL Vertebrate Level). The AGL locality is the western-most locality in the Sacaco area (*Ochoa et al., 2021*: Fig. 3), and the sequence has at least five different fossil-bearing horizons distributed over about 200 m of sediment. The AGL Vertebrate Level extends between about 35–68 m from the base of *Ochoa et al.*'s *(2022)* Section S1, and it contains *Brand et al.*'s *(2011)* guide beds LM-10 and LM-11. The base of the AGL Vertebrate Level is constrained by a U-Pb zircon dating (sample MG-63) collected ~9 m from the base of the Section S1 providing a weighted mean lower age of 9.2 ± 0.08 Ma. *Muizon & Bellon* (*1986*: 1403) reported two K-Ar ages based on biotite (samples 45 and 44) at 8.04 ± 0.40 Ma and 7.93 ± 0.40 Ma (respectively), which were presumably collected from the AGL Vertebrate Level. Currently, the best estimation for the stratigraphic position of these samples places them below the fossil-bearing horizon (*Lambert & De Muizon, 2013*; see *Ochoa et al., 2022*). We note that (*Muizon et al., 2003*), in their original description, reported the age of *T. antiquus* as Huayquerian (South American Land Mammal Age, ca. 8 Ma). An upper age constraint for the AGL Vertebrate Level derives from diatom biostratigraphy of the Cerro Vildoso horizon in sequence at the AGL locality, located about 62 m above the base of Section S2 in *Ochoa et al. (2022)*. Accordingly, based on radiometric dates and diatom biostratigraphy, *T. antiquus* is Tortornian-Messinian in age from 9.2 to ~5.6 Ma.

The type specimen of *T. natans* (MNHN SAS 734) was collected from the Montemar locality from strata with a fossil-bearing horizon of the same name, reported as ~7.3 Ma by *Ochoa et al. (2021)*, which referred to the *Ehret et al.*'s *(2012)* median age of ~7.3 Ma (CI [8.7–6.45 Ma]) using strontium from the MTM fossil-bearing level at the locality. Later, *Ochoa et al. (2022)* recalculated this value as ~7.15 Ma (CI [8.25–6.52 Ma]). Also,

**Table 8** *Thalassocnus* localities at the Pisco Formation. Published specimens for type species of *Thalassocnus* from Pisco Formation localities in Peru.

| Taxon | Type specimen | Locality | Horizon | Reference for type locality and horizon | Age |
|---|---|---|---|---|---|
| *T. yaucensis* | MUSM 37 | Yauca Depressions, Sacaco area | "Probably slightly younger than the SAO Horizon" | *Muizon et al. (2004a)*: 389 | 5.35–4.8 Ma |
| *T. carolomartini* | SMNK PAL 3814 | Sacaco Chacra, Sacaco area | SAO horizon | *Muizon et al. (2004a)*: 389 | 5.9–~5.7 Ma |
| *T. littoralis* | MNHN SAS 1615 | Sud-Sacaco West, Sacaco area | SAS horizon | *McDonald & De Muizon (2002)*: 356 | 6.28-5.6 Ma |
| *T. natans* | MNHN SAS 734 | Montemar, Sacaco area | MTM horizon | Locality: *Muizon & McDonald (1995)*: 224; Horizon: *McDonald & De Muizon (2002)*: 351 | ~7.15–6.3 Ma |
| *T. antiquus* | MUSM 228 | Aguada de Lomas, Sacaco area | AGL vertebrate level | Locality: *Muizon et al. (2003)*: 886; Horizon: *Muizon et al. (2003)*: 888. | 9.2–~5.6 Ma |

**Note:**
Horizon, locality and age data are revised from the original publication as described in the main text. Abbreviations: MNHN, Muséum national d'Histoire naturelle, Paris, France; MUSM, Museo de Historia Natural de la Universidad Nacional Mayor de San Marcos, Lima, Peru; SMNK, Staatliche Museum für Naturkunde, Karlsruhe, Germany.

*Ochoa et al. (2022)* reported a U-Pb weighted mean age of 6.33 ± 0.03 Ma for a tuff sample (sample MG4-06) below the erosive surface the top of this section (S5), which is located below a Mio-Pliocene unconformity identified by *Muizon & DeVries (1985)*, thereby constraining the MTM horizon and age of *T. natans* as Tortonian-Messinian from ~7.15 to 6.3 Ma.

*McDonald & De Muizon (2002)* reported that the type of *T. littoralis* (MNHN SAS 1615) was similarly collected from a fossil-bearing horizon and a locality of the same name: Sud-Sacaco (SAS). We note that (*Muizon & DeVries, 1985*) identified two different localities bearing this name: Sud-Sacaco (West), which is located west of the Pan-American Highway; and Sud-Sacaco (East), located on the east side of the highway but south of Sacaco, south of the Acari River. *Ochoa et al. (2022)* conducted measurements of sections S5 and S6, which correspond to the same areas previously studied by *Muizon & DeVries (1985)*. Section S5 includes the upper portion of the section from the MTM locality, while section S6 is identified as Sud-Sacaco West. Both sections are situated west of the Pan-American Highway and are less than 1 kilometer apart. The preponderance of articles describing taxa from the SAS horizon are presumably from these sequences (*i.e.*, Sud-Sacaco West). Based on radiometric data and diatom biostratigraphy, *Ochoa et al. (2022)* argued for a lower depositional age of 6.28 Ma and an upper age of 5.6 Ma, which differs slightly from the corrected (*Ehret et al., 2012*) strontium values of 6.55 Ma (CI [7.2–6.1 Ma]) and 5.9 Ma (CI [6.25–5.52 Ma]) presumably for the SAS fossil-bearing horizon. *Ochoa et al. (2022)* made it clear that the oldest age of 6.28 ± 0.05 Ma derives from a sample tied to the levels of the Sud-Sacaco West sequence that overlie the MTM sequence (MG4-25); strontium dates from two shark teeth (SAC 20–29-T1, -T2) collected at the top of section S6 yield dates of 5.85 and 6.15 Ma. Thus, the best age estimate for *T. littoralis* is Messinian at 6.28-5.6 Ma. *Ochoa et al. (2021*: S13) noted *T. littoralis* from Aguada de

Lomas at LM15 (*Brand et al., 2011*), which is located towards the top of section S2 in *Ochoa et al. (2022)* at 50 m above the base, about 12 m below the Cerro Vildoso horizon. However, the top of the AGL sequence has a similar age to the top of the Sud-Sacaco West sequence.

The type specimen of *T. carolomartini* (SMNK PAL 3814) was collected from the Sacaco (SAO) locality, which has the most prominent fossil-bearing horizons in the entire Pisco Formation also using the same name, the SAO horizon. This locality is north of the Panamerican Highway, close to the Paleontological Museum in the town of Sacaco. Here we follow *Ochoa et al. (2022)*, who renamed the SAO locality as Sacaco Chacra. Based on a variety of radiometric dates presented by previous studies and new strontium ages based on shark teeth, *Ochoa et al. (2022)* concluded that the fossil-bearing sedimentary rocks at Sacaco Chacra are Messinian from 5.9 to ~5.7 Ma, including the age of *T. carolomartini*.

The youngest sedimentary rocks bearing marine vertebrate fossils in the Sacaco area are located further south along the Panamerican Highway near the town of Yauca. In their description of the type of *T. yaucensis* (MUSM 37), *Muizon et al.* (*2004a*:389) argued that it was "probably slightly younger than the SAO Horizon" from the type locality of East Yauca. *Ochoa et al. (2022)* clarified that this locality is the Yauca Depressions (their section S8), located a few hundred meters southwest of the Panamerican Highway, west of the town of Yauca. The Yauca Road Cut in the town of Yauca represents older Tortonian sediments. *Ochoa et al. (2022)* reported strontium dates from shark teeth in section S8 from the Yauca Depressions with a lower age of 5.35 ± 0.15 Ma and an upper age of 4.85 ± 0.15 Ma. Additional radiometric dates limited a maximum depositional age of 4.8 Ma for the Yauca Depressions locality. Thus, *T. yaucensis* is nearly entirely Zanclean in age ~5.35-4.8 Ma.

*Bahía Inglesa Formation*–A partial and isolated right dentary (SGO.PV 1093) was reported from the locality Estanques de Copec (which we argue is equivalent to or, minimally, at the same stratigraphic level as the specimens from NBC described herein). The morphology of the dentary (*e.g.*, distinct concavo-convex ventral profile, dorsolaterally directed opening of the mandibular canal, and m2 with high and nearly straight transverse crests) and its overall size are comparable to *T. antiquus* and *T. natans*, preventing its assignation to either species (*Canto et al., 2008*; Table 1). Records of aquatic sloths from this formation also comprise left and right femora (SGO.PV 1133) from the Arenas de Caldera locality (Fig. 1B). The biochronology of invertebrate species suggests that the fossil-rich layers at this locality range from the middle Miocene to the early Pliocene (*Guicharrousse-Vargas et al., 2021*). The specimen SGO.PV 1133 was preliminarily referred to *Thalassocnus* sp.; however, no descriptions of their comparative morphology have been provided (*Suárez et al., 2011*). Nevertheless, this specimen is nearly identical to MPC 705-A, but slightly shorter and more robust than MPC 704-A (J Velez-Juarbe, 2012, personal observations) Additional *Thalassocnus* remains have been recovered from the nearby Parque Paleontológico Los Dedos locality (Late Miocene) south of Caldera (Fig. 1B). This occurrence includes a distal fragment of a right humerus (CPUC/C/557), which was

tentatively identified as *Thalassocnus* cf. *T. natans* based on the shape of the region between the capitulum and the trochlea and the distal edge of the trochlea (*Peralta-Prato & Solórzano, 2019*).

*Thalassocnus* remains have also been excavated from the BL-4 horizon at the Cerro Ballena locality (~6.1 Ma; *Martinez et al., 2025*), about 1.5 km northeast of NBC (Fig. 1B) (*Pyenson et al., 2014*). These remains include an isolated right femur (MPC 644) and the posterior section of a left horizontal ramus (MPC 704). MPC 644 was identified as *T. natans* based on the presence of an elongated shelf formed by the greater and third trochanter, the shape and position of the fovea capitis, and the continuous surface between the patellar surface and the condyles. Although MPC 704 did not preserve diagnostic traits, it was excavated from the same level as MPC 644, which led the authors to identify it as *T. natans*. *Amson et al. (2015b)* discussed that MPC 644 and 704 lack diagnostic features at the species level and that the morphology of MPC 644 resembles that of both *T. natans* and *T. antiquus*. However, additional qualitative and quantitative comparisons support the identification of MPC 644 as *T. natans* (see discussion above). Still, we re-identify MPC 704 as Folivora indet. pending further material collection from this site.

*Coquimbo Formation*–This formation comprises an extensive sequence of terraces featuring neritic or sublittoral marine sediments with abundant fossil fauna ranging from the Middle Miocene to the Late Pliocene from north-central Chile (*Le Roux et al., 2005*; *Acosta-Hospitaleche, Canto & Tambussi, 2006*; *Chávez Hoffmeister, 2007*; *Salinas, 2011*; *Staig et al., 2015*; *Partarrieu et al., 2018*, *2025*). From this formation a partial postcranial skeleton of *Thalassocnus* (SGO.PV 15500) was excavated in a fossil-rich Pliocene level at Lomas del Sauce locality (Fig. 1D) (*De Los Arcos et al., 2017*; Table 1). The specimen, including nearly complete fore- and hind limbs, vertebrae and some ribs, display morphological characteristics intermediate between *T. carolomartini* and *T. yaucensis*. Although the primary diagnostic traits distinguishing these two species are located on the skull, SGO.PV 15500 was tentatively identified as *T. carolomartini* based on characteristics of the humerus (*e.g.*, symmetrical widening of the distal end), ulna (*e.g.*, the presence of a sulcus extending from the central groove for the extensor carpi radialis muscle to the articular facet with the carpus), and astragalus (*e.g.*, the angle formed between the odontoid and discoid trochlea) (*De Los Arcos et al., 2017*). Further research on *Thalassocnus'* postcranial morphology, both qualitatively and quantitatively, is essential for verifying this identification. Still, with the available information, SGO.PV 15500 marks the southernmost occurrence of *T. carolomartini* in South America and the first in Chile.

*Horcón Formation*–Fossiliferous levels from the Horcón Formation along the coastline of central Chile have yielded abundant remains of mollusks, and marine vertebrates, such as sharks, sea birds, seals, and cetaceans (*e.g.*, *Chávez Hoffmeister, Carrillo Briceño & Nielsen, 2014*; *Benites-Palomino et al., 2022*). Although no specific dating exists for this formation, the invertebrate fauna constrains the age of the Horcón Formation to the upper Pliocene (*DeVries, 1997*, *2003*). The only *Thalassocnus* specimen (SGO.PV 21545) from the Horcón

Formation is from the Playa La Luna locality (Fig. 1E) (Table 1), and consists of an isolated co-ossified right proximal and middle phalanges of the third digit of the pes (*De Los Arcos et al., 2017*). The specimen shares certain resemblances (*e.g.*, the depth of the plantar fossa and proximodistal length) with chronostratigraphic species younger species than *T. natans* (*i.e.*, *T. littoralis*, *T. carolomartini*, and *T. yaucensis*). Its specific identity remains uncertain until additional material from this formation is obtained (*De Los Arcos et al., 2017*).

*Remarks*–Several remains of *Thalassocnus* have been reported from various late Neogene fossil-bearing deposits in southern Peru and northern and central Chile (Fig. 1 and Table 1). These records have uncovered intriguing aspects of the secondary adaptation to the marine environment among tetrapods and the evolution of the marine ecosystems (see below). However, uncertainties about the stratigraphic origin and age associated with *Thalassocnus* remains have resulted in doubtful interpretations and an overstatement of their biostratigraphic and biochronological suitability (Table 1; see *De Los Arcos et al., 2017*). Recent studies have greatly improved the geological context and dating of fossil-rich horizons and localities in the Bahía Inglesa and Pisco formations (*e.g.*, *Ochoa et al., 2021*; *Martinez et al., 2025*), which better constrain some of the geologic age estimates.

This revised chronostratigraphic framework directly impacts our understanding of the *Thalassocnus* lineage. The revised ages of the fossiliferous levels from the Pisco Formation show that *T. antiquus* was coeval with both *T. natans* and *T. littoralis*, challenging the hypothesis of *Thalassocnus* evolution through an anagenetic succession. Additionally, these new estimates indicate that the *Thalassocnus* remains from the upper Pliocene Horcón Formation at the Playa La Luna locality in Chile represent the youngest known records of this genus to date.

Although nearly contemporaneous, the *Thalassocnus* fossil record from both Chile and Peru shows significant distinctions. Historically, published records from Peru have centered on describing type specimens, which comprise nearly complete skeletons that offer unparalleled detailed information about their osteology. However, this practice has also led to some species being known from single but nearly complete skeletons (*e.g.*, *T. yaucensis*), leaving open questions about intraspecific variability, including sexual dimorphism. Conversely, *Thalassocnus* records from Chile mainly comprise various occurrences from different formations, primarily consisting of incomplete, isolated, or associated postcranial remains that possess few diagnostic traits. This limitation hinders species identification and prevents a comprehensive examination of their local taxonomic diversity and other aspects of their paleobiogeography and paleoecology. Nevertheless, the study of the Chilean *Thalassocnus* record has uncovered some elements that deserve further attention. Indeed, some appendicular remains from the Bahía Inglesa and Coquimbo formations exhibit intermediate morphological characteristics or a combination of traits previously described for certain species (*e.g.*, SGO.PV 15500, SGO.PV 21545). These features indicate variability in the genus *Thalassocnus* and underscore the necessity to further investigate the postcranial morphology of aquatic sloths.

## Disparities in marine herbivore guilds across the Pacific

The Pacific Ocean coastlines have fostered the evolution of various lineages of herbivorous mammals throughout the Cenozoic (*Steneck, Bellwood & Hay, 2017*). In the North Pacific, herbivorous mammals consistently occupied the grazing niche from the Oligocene to the Holocene, starting with the evolution of desmostylians and, more recently, the dispersal of sirenians from the Atlantic (*Vermeij, 2018*; *Vermeij et al., 2019*). Desmostylians were an extinct group of herbivorous, hippo-sized, quadrupedal aquatic mammals that evolved in the northeastern Pacific during the early Oligocene (*Matsui, Valenzuela-Toro & Pyenson, 2022*). The extensive fossil record indicates that they likely grazed on seagrasses and other forms of aquatic vegetation while inhabiting estuarine and coastal freshwater systems (*Clementz, Hoppe & Koch, 2003*; *Valenzuela-Toro et al., 2024*). These animals varied in body size and inhabited the eastern and western coasts of the North Pacific until the Late Miocene. They likely became extinct due to competition for marine vegetation with the diversification of sirenians, with whom they significantly overlapped in both time and space (*Velez-Juarbe, Domning & Pyenson, 2012*; *Vermeij, 2018*). After the disappearance of desmostylians, diverse communities of dugongid sirenians dominated the North Pacific grazing niche until the extinction of Steller's sea cows (*Hydrodamalis gigas*) in the eighteenth century, marking the end of large herbivorous mammals in the North Pacific (*Aranda-Manteca, Domning & Barnes, 1994*; *Domning & Furusawa, 1994*; *Pyenson & Vermeij, 2016*).

Less is known about the evolution of herbivorous aquatic mammals and grazing along the South Pacific coastlines in the geologic past (*Pyenson & Vermeij, 2016*). Still, they were likely influenced by the rise of seagrass-dominated ecosystems in southern South America during the early Miocene (*Panti et al., 2025*) and by a warmer, wetter climate in the late Miocene (*Ochoa et al., 2025*). The oldest records of herbivorous mammals from this region consist of cranial remains of the dugongid *Metaxytherium crataegense* from the Early Miocene of Peru (*Muizon & Domning, 1985*). The fossil record suggest they were the exclusive members of marine herbivore guilds in the South Pacific until the Late Miocene-Pliocene, when a group of dwarf dugongines (*Nanosiren* spp.) and aquatic sloths (*Thalassocnus* spp.) evolved in this region (*Viglino et al., 2023*). Nevertheless, records of *Nanosiren* are rare, and correspond to isolated remains from Pisco and Bahía Inglesa formations (*Muizon & Domning, 1985*; *Bianucci et al., 2006*; *Domning & Aguilera, 2008*). Although no direct evidence of their paleoecology exists, the small body size (~2 m, ~116 kg; *Sarko et al., 2010*), along with other morphological traits observed in related specimens from the Western Atlantic and Caribbean, indicates that *Nanosiren* likely fed on small seagrasses in shallow waters (*Domning & Aguilera, 2008*; *Velez-Juarbe, Domning & Pyenson, 2012*). Similarly, morphological studies of chronostratigraphic older aquatic sloth species (*i.e.*, *T. antiquus*, *T. natans*, *T. littoralis*) suggest they likely foraged on shallow seagrasses, whereas, presumably younger species (*i.e.*, *T. carolomartini* and *T. yaucensis*) may have been more specialized grazers in deeper habitats (*Muizon et al., 2004b*). The causes of the extinction of herbivorous aquatic mammals in the South Pacific remain

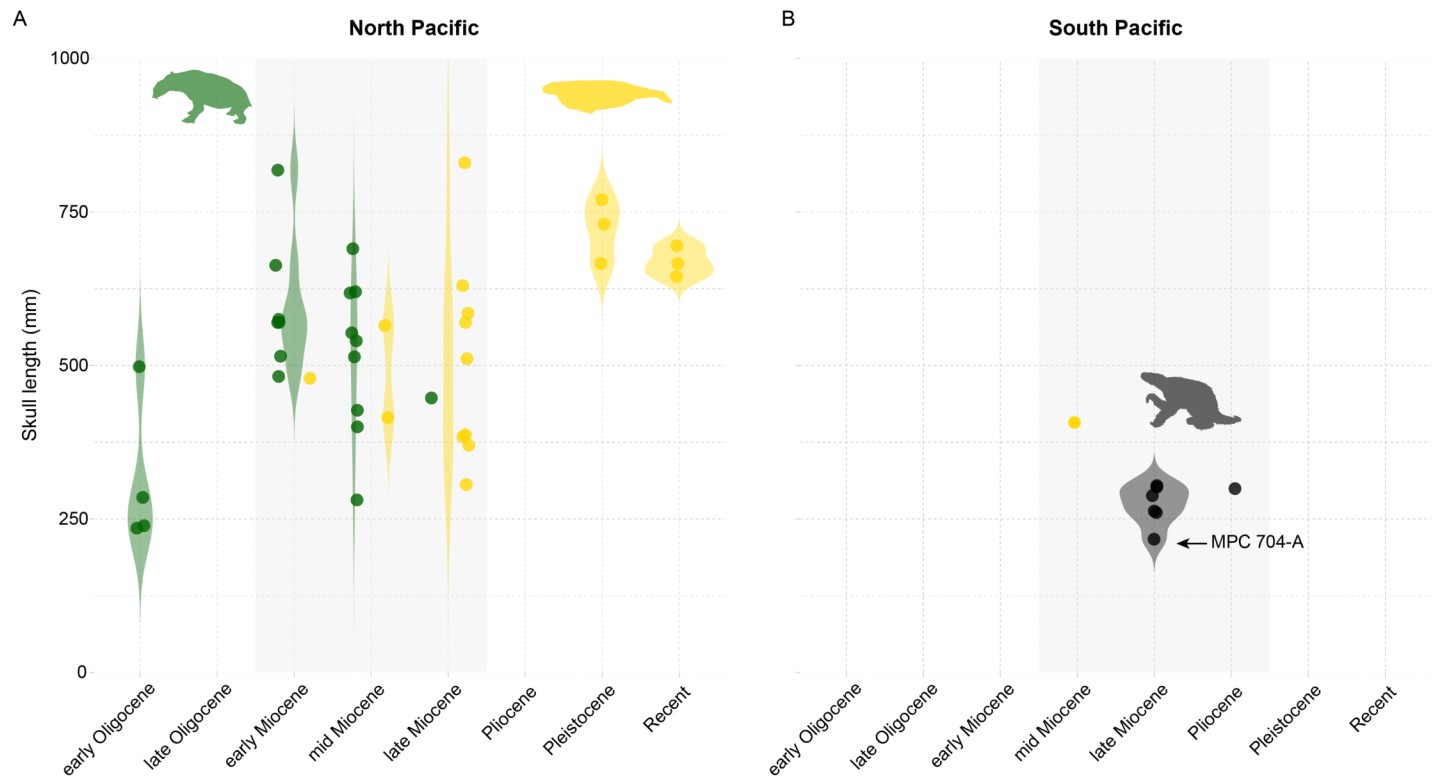

**Figure 16 Variations in body size among groups of marine herbivores throughout the North (A) and South (B) Pacific during the Cenozoic.** Violin plots of measurements of skull length (an indicator of body length in mammals) for Desmostylia (green), stem and crown Sirenia (yellow), and *Thalassocnus* (black) during the Cenozoic—Desmostylian and Sirenian data sourced from *Pyenson & Vermeij (2016)*. Refer to Table S6 for details. Animal silhouettes are not to scale and were obtained from Phylopic (phylopic.org).

unclear. Nevertheless, it has been speculated that the disappearance of sea grasses during the Pliocene in this region could have played a role (*Vermeij, 2018*).

Although herbivorous guilds of aquatic mammals in the Pacific exhibit some similarities, such as South Pacific aquatic sloths serving as ecological counterparts to North Pacific desmostylids (*i.e.*, high-crowned herbivores), the differences among these guilds are extensive. Records in the South Pacific are chronostratigraphically younger and less taxonomically diverse than their North Pacific counterparts. Notably, herbivorous aquatic mammals from the South Pacific also attained body sizes (skull length: ~22–47 cm) within the range of some of the smallest desmostylians (skull length: ~23.5 cm) and were smaller than most sirenians (skull length: ~19.5–83 cm) from the North Pacific (Figs. 16A, 16B). During the Late Miocene, which is the timeframe for which size data of contemporaneous marine herbivores from the North and South Pacific are available (except for a Middle Miocene sirenian record from Peru), significant differences in body size are observed (Kruskal-Wallis test: $\chi^2(2) = 10.6$, $p = 0.0050$). Specifically, *Thalassocnus* had significantly smaller sizes than contemporaneous North Pacific sirenians (Wilcoxon rank sum test: $p = 0.0020$).

The drivers of these disparities remain unknown (*Pyenson & Vermeij, 2016*). Nonetheless, the maximum body size of consumers reflects the availability of resources,

with size differences within a guild likely reflecting variations in the productivity of a specific category of resources over geologic time (*Vermeij, 2011*). We hypothesize that size variations among marine herbivore guilds might have resulted from regional-scale differences in productivity on the seafloor during the Neogene across the Pacific. The North Pacific had higher productivity and greater availability of food resources (*e.g.*, aquatic vegetation) than the South Pacific during the Neogene, enabling marine herbivorous mammals to reach larger sizes.

The fossil record of marine vegetation provides some support to our hypothesis. Fossils suggest that the North Pacific fostered the early evolution of kelps (brown macroalgae belonging to the order Laminariales) during the Oligocene (*Kiel et al., 2024*). Kelps are considered ecosystem engineers and form diverse multi-layered underwater beds in subtidal and intertidal zones (*Teagle et al., 2017*). They are a major contributor to coastal productivity, enhancing energy and nutrient cycling by direct herbivory and boosting detrital production (*Mann, 1973*; *Teagle et al., 2017*). Fossils suggest that kelp beds existed in the North Pacific since the beginning of the Neogene, likely enhancing productivity and providing resources for ancient coastal communities, including early grazers (*Vermeij, 2018*). Stable isotope analyses suggest desmostylians foraged on seagrasses (*Clementz, Hoppe & Koch, 2003*; *Valenzuela-Toro et al., 2024*), although kelp consumption remains a possibility that should be investigated. Cenozoic sirenians in the North Pacific also exhibited a variety of dietary specializations, ranging from seagrasses and rhizomes to kelp (*Aranda-Manteca, Domning & Barnes, 1994*; *Velez-Juarbe, 2014*). Even if kelp did not directly contribute to the diet of sirenians and desmostylians, they may have benefited indirectly. Ancient kelp forests would have had cascading effects throughout the Cenozoic food webs, supporting the proliferation of understory vegetation, much like modern marine ecosystems (*e.g.*, *Duggins, Simenstad & Estes, 1989*; *Page et al., 2008*; *Krumhansl & Scheibling, 2012*; *Elliott Smith, Harrod & Newsome, 2018*), and ultimately favoring marine herbivores evolving larger body sizes, especially multispecies herbivore assemblages that lower interspecific competition (*Velez-Juarbe, Domning & Pyenson, 2012*).

Kelp assemblages have a different evolutionary biogeography in the South Pacific (*Bolton, 2010*), likely affecting marine herbivore guilds. Here, brown macroalgae encompass representatives of the Laminariales and the genus *Durvillaea*, exclusive to the Southern Hemisphere. These two groups boost marine productivity but they lead to varied effects on ecosystems. Laminariales primarily support subtidal ecosystem productivity and enhance large-scale coastal nutrient dynamics, similar to their northern counterparts (*Graham, Halpern & Carr, 2008*). Conversely, members of *Durvillaea* are morphologically robust algae that thrives in high-energy intertidal zones, aiding offshore nutrient dispersal and influencing long distance productivity (*Fraser et al., 2020*). While species of *Durvillaea* originated in the southwestern Pacific during the Oligocene and early Miocene (*Fraser et al., 2010*; *Vermeij et al., 2024*), representatives of the Laminariales evolved from North Pacific ancestors, colonizing the Southern Hemisphere much later, during the Pliocene (*Jackson et al., 2017*). Therefore, although the South Pacific coastlines were likely productive during the Neogene (*e.g.*, *Ochoa et al., 2021*), subtidal and intertidal production may have been relatively lower than in the North Pacific due to differences in the marine

vegetation, contributing to the smaller body sizes observed among marine herbivores in the South Pacific.

An alternative but complementary explanation for the observed body size disparity is also represented by the divergent digestive physiologies of these herbivore groups. As inferred non-ruminant foregut fermenters based on their phylogenetic relationship to modern and extinct relatives (*e.g.*, *Tejada-Lara et al., 2018*), *Thalassocnus* would have maximized nutrient extraction from low-quality forage. However, this high-efficiency strategy is constrained by a more extended ingesta retention time, which imposes a limit on additional energy gain and food intake (*Clauss et al., 2003*, *2007*). In contrast, tethytheres such as sirenians and desmostylians were hindgut fermenters (*Best, 1981*; *Reynolds & Rommel, 1996*). While the hindgut fermentation strategy is generally less efficient per unit of food compared to foregut fermentation, it permits a more rapid processing rate, allowing animals to fuel their metabolism by increasing total food consumption (*Clauss et al., 2003*), with some groups like sirenians evolving a particularly high overall digestive efficiency for cellulose within this system (*Burn, 1986*). The consequences of these physiological differences were likely profound. While an internal metabolic ceiling likely constrained the maximum size of *Thalassocnus*, the growth of sirenians and desmostylians was primarily limited by food availability. In this context, the predicted high primary productivity of the North Pacific, driven by the early evolution of tidal and subtidal kelp forests, would set a favorable scenario, providing the resources necessary for hindgut-fermenting sirenians and desmostylians to fulfill their higher food requirements, allowing them to reach larger sizes relative to the physiologically-constrained *Thalassocnus* in the South Pacific.

Biases within the fossil record could also contribute to, or at least amplify, the observed disparity in body size among marine herbivores across the Pacific. Taphonomic processes, for instance, can favor the preservation of larger, more robust bones over smaller, more fragile remains (*e.g.*, *Brown et al., 2022*). An asymmetrical research effort can exacerbate this physical bias, as the North Pacific has a longer tradition of paleontological investigation with a greater number of researchers (*e.g.*, *Valenzuela-Toro & Pyenson, 2019*), making it more likely that the largest individuals of the northern faunas have been uncovered and published, potentially exaggerating any true biological size differences between the two regions. Future research should evaluate these factors and contribute to disentangling the evolutionary and ecological trajectories of coastal ecosystems across the Pacific over geologic time.

## CONCLUSIONS

We report a new, nearly complete skeleton of the aquatic sloth *Thalassocnus natans*, accompanied by associated cranial and postcranial remains, from the Late Miocene Norte Bahía Caldera locality within the Bahía Inglesa Formation (Atacama region, Chile). Representing the most complete *Thalassocnus* material found in Chile to date, these specimens, excavated from likely Late Miocene fossiliferous strata at locality NBC, exhibit morphology consistent with previously described *T. natans* remains. However, observed

morphological differences in the femora and tibiae may indicate intraspecific variation or sexual dimorphism. A review of the published geological age estimates for *Thalassocnus*-bearing deposits from the Pisco Formation, along with records of this genus from Chile, indicates that the *Thalassocnus* remains from the upper Pliocene of central Chile represent the youngest record of this genus reported to date. The new fossil material from NBC, combined with other *Thalassocnus* finds and records of South American marine mammal herbivores (*e.g.*, sirenians) from Chile and Peru, supports our conclusion that these South American herbivores did not reach the large body sizes of their North Pacific ecological counterparts, the desmostylians and sirenians. We propose that this size disparity likely arose from regional-scale differences in Neogene seafloor productivity across the Pacific, with the higher productivity and greater availability of aquatic vegetation in the North Pacific potentially enabling the evolution of larger marine herbivorous mammals compared to the South Pacific.

## INSTITUTIONAL ABBREVIATIONS

**MNHN**    Muséum National d'Histoire Naturelle, Paris, France
**MPC**     Museo Paleontológico de Caldera, Caldera, Chile
**MUSM**    Museo de Historia Natural, Universidad Nacional Mayor de San Marcos, Lima, Peru
**SMNK**    Staatliche Museum für Naturkunde, Karlsruhe, Germany
**SGO.PV**  Museo Nacional de Historia Natural, Santiago, Chile

## ACKNOWLEDGEMENTS

We express our recognition to M. Chávez Hoffmeister and C. Simon Gutstein for providing comments and suggestions that improved preliminary versions of the manuscript. Ana M. Valenzuela-Toro thanks the Gobierno Regional de Atacama for its support during this project. We extend our gratitude to R. Salas-Gismondi and R. Varas (MUSM), C. Argot, G. Billet, and C. de Muizon (MNHN), and R. Figueroa and M. Forch (MPC) for access to collections under their care and I. Tapia (CIAHN Atacama) for technical assistance during the final phase of the study. We are also grateful to Philip Reno and an anonymous reviewer and R. Salas-Gismondi (UPCH) for their careful and constructive review of our manuscript, and to the latter for providing photos of Thalassocnus littoralis which facilitated our comparisons. Ana M. Valenzuela-Toro acknowledges the use of Grammarly to assist with grammar and spelling corrections. This is Caldera Paleontology Project contribution No. 3. This study is dedicated to the memory of James A. Estes.

### Funding

During the performance of this study Ana M. Valenzuela-Toro was funded by ANID PCHA/Becas Chile, Doctoral Fellowship (Grant No. 2016-72170286), and a Peter Buck

Predoctoral Fellowship by the National Museum of Natural History Smithsonian Institution, whereas Jorge Velez-Juarbe was funded by a predoctoral fellowship from the National Museum of Natural History, Smithsonian Institution and National Science Foundation Earth Sciences Postdoctoral Fellowship #1249920. Writing for this manuscript was also funded by a NMNH Small Grant Award, discretionary funding from NMNH Office of the Director, the Smithsonian Institution's Remington Kellogg Fund, two National Geographic Society Committee on Research Exploration grants (8903-11, 9019-11) to Nicholas D. Pyenson, and by U-REDES (Línea Domeyko 2 UR-C12/1, Universidad de Chile). The funders had no role in study design, data collection and analysis, decision to publish, or preparation of the manuscript.

### Grant Disclosures
The following grant information was disclosed by the authors:
ANID PCHA/Becas Chile: 2016-72170286.
National Museum of Natural History, Smithsonian Institution.
National Science Foundation Earth Sciences Postdoctoral Fellowship: #1249920.
NMNH Small Grant Award.
NMNH Office of the Director.
Smithsonian Institution's Remington Kellogg Fund.
National Geographic Society Committee on Research Exploration: 8903-11, 9019-11.
U-REDES (Línea Domeyko 2 UR-C12/1, Universidad de Chile).

### Competing Interests
Nicholas D. Pyenson is an Academic Editor for PeerJ. Mario E. Suárez is an employee of Atacama Fósil Limitada.

### Author Contributions
- Ana M. Valenzuela-Toro conceived and designed the experiments, performed the experiments, analyzed the data, prepared figures and/or tables, authored or reviewed drafts of the article, and approved the final draft.
- Nicholas D. Pyenson conceived and designed the experiments, performed the experiments, analyzed the data, prepared figures and/or tables, authored or reviewed drafts of the article, and approved the final draft.
- Jorge Velez-Juarbe conceived and designed the experiments, performed the experiments, prepared figures and/or tables, authored or reviewed drafts of the article, and approved the final draft.
- Mario E. Suárez conceived and designed the experiments, authored or reviewed drafts of the article, excavated the fossils, and approved the final draft.

### Data Availability
Raw morphological measurements of the fossil specimens are available in Tables 1–8 and the Supplemental Files.

## Supplemental Information

Supplemental information for this article can be found online at http://dx.doi.org/10.7717/peerj.19897#supplemental-information.

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
