# Peer review of "Aquatic sloths (Thalassocnus) from the Miocene of Chile and the evolution of marine mammal herbivory in the Pacific Ocean"

_PeerJ, doi:10.7717/peerj.19897_

## Round 0.1 · original submission · Minor Revisions

We are resending the Decision for your article in an attempt to address the error you are receiving then you try to resubmit.


We received detailed reviews from two experts who are excited and complimentary about your work. They each provided detailed comments and revisions that struck very similar themes. They both classified these as minor revisions, but they acknowledge they are broad and necessary. These suggestions extend to data inclusion and interpreation and require the editing and addition of figures. These seem potentially extensive to me, but I trust their opinion and have classified the manuscript as needing Minor Revisions.

However, both authors provide comments regarding the evidence and interpretation of the species designation and the implications for the temporal setting. They also provide comments and require further discussion regarding your ecological interpretations. In addition, numerous smaller, targeted changes are indicated in the review and the attached comments to the manuscript.

Please revise your analysis and manuscript accordingly. I look forward to receiving your revised manuscript and author response to this intriguing analysis.

Reviewer 1 ·

Basic reporting

Manuscript is very well written, although a few misspellings were encountered - they are noted on the review pdf

Experimental design

no comment

Validity of the findings

Regarding conclusions, the manuscript spends a decent amount of time on the geological context for Thalassocnus, which is reasonable given the establishment of the five recognized species into geochronological bins. However it seems the new specimens morphologically fall within a species (T. natans) but not in that species' timespan - and very little is said about this. Either I have missed something regarding the localities and dates, or an expanded temporal range for this species is being undervalued for its implications.
If it isn't an expansion, then I would question the necessity of including so much of the geologic and temporal data in the manuscript.

On another note, I would advise that the measurements of foramens be removed from the manuscript text. These are features of variability due to their nature in individuals and are not dependable for taxonomic/diagnostic means. As such, their inclusion serves no real purpose. A similar argument could also be made regarding some of the included measurements of various entheses.

Additional comments

The annotated pdf covers some of the various comments left above but addresses a few additional things that would help to further strengthen the manuscript. One of note is the incorrect identification of the iliopectineal eminence on the pelvis.

Annotated reviews are not available for download in order to protect the identity of reviewers who chose to remain anonymous.

·

Basic reporting

The manuscript describes in detail new fossil material of the aquatic sloth Thalassocnus from the Late Miocene of Bahía Inglesa, northern Chile. In addition, the authors conduct a chronological review of the localities that have provided Thalassocnus material from Peru and Chile. Finally, based on the fossil record, the differences between Northern and Southern hemisphere guilds of Neogene aquatic herbivores are discussed.
The English language is adequate, and the ideas are clearly organized. The fossil material includes the most complete Thalassocnus specimen discovered in Chile, which preserves a partial skull and postcranial elements. Additionally, a partial dentary and long bones of a hind limb belonging to different individual are described. All these specimens were recovered from a new locality in Bahía Inglesa, called Norte Bahía de Caldera. The rocks from this locality are considered to belong to the Late Miocene of the Bahía Inglesa Formation.
References are adequate and up-to-date. The introduction is adequate as well and provides relevant data/background for a correct understanding of the manuscript. The overall structure of the manuscript is well-suited to the way the topic is developed. The figures and tables are well-organized and accurately reflect the raw data; however, I suggest including some additional figures (see below).
The approach of this manuscript is primarily descriptive and qualitative. The description of the fossil remains are rigorous and detailed. Based on the comparative anatomical description, it is concluded that the specimens identified at the species level belong to Thalassocnus natans, one of the five species previously described from fossil material found in the rich paleontological locality of Sacaco, Peru.
While the manuscript is an important contribution to the understanding of Thalassocnus diversification in the southern Pacific, the manuscript's two main conclusions—the specific identity of the new Thalassocnus material (partially) and the differences between northern and southern marine herbivore guilds during the Neogene—require further evaluation and additional deeper/quantitative analyses to be supported. I detail my observations and suggestions in the following sections. This should be improved upon before acceptance. Although these might be considered substantial changes, they are just minor adjustments that won't be difficult to implement.

Experimental design

Regarding the experimental approach, as I mentioned, this is a study based on the anatomical, qualitative description of new Thalassocnus remains from Chile and a review of Thalassocnus records and their ages from Peru and Chile. The first main goal of this research- the taxonomic identification of the Chilean specimens reported here – could benefit from additional comparisons, linear morphometry, and some figures that could be included. The second main goal – differences between northern and southern marine herbivore guilds during the Neogene – requires further analyses.

That said, these are my comments and suggestions:
- Taxonomic identity of the material from Norte Bahía de Caldera. Specimen MPC 704-A consists of a partial skull and postcranial elements. Diagnostic elements, such as the radius, ulna and skull resembles closer to Thalassocnus natans than other species recognized from Peru. However, the differences it presents with the holotype of T antiquus are not notable. Because (1) radii, ulnae and cranial elements of all Thalassocnus species exist and (2) the differences between species are mostly based on variation in forelimb bone proportions, I suggest performing linear (traditional) morphometric analyses that allow a more precise visualization of its morphology in relation to the other species. I also suggest a figure comparing radii and ulnae of Thalassocnus species from Peru and Chile.

Specimen MPC 705 has also been identified as T. natans. However, its anatomy and proportions are closer to those of MUSM 223 (Vertebrate Paleontology Department in Natural History Museum in Lima, Peru), a specimen from Peru identified as T. littoralis (in the manuscript; Amson 2015a, 2015b, 2015c; incorrectly referred to as T. natans in Salas et al., 2005), and to the holotype of T. carolomartini. The postcranial material of MUSM 223 has been figured and published but the dentary is not figured yet in scientific literature. The length of the predentary region of MPC 705 is observed to be similar to that of MUSM 223. In T. natans is much shorter. Furthermore, the shape of the spout tip is not downward sloping, and the ventral profile of the spout is not concave, similar to that preserved in younger Thalassocnus species, and in contrast to T. natans, T. antiquus, and even T. littoralis. In MPC 705, the posterior end of the symphysis (in dorsal view) and the anterior edge of the base of the coronoid process (in lateral view) are located away from the dental series, as in MUSM 223, unlike what is observed in other Thalassocnus species. If necessary I can provide images of the dentary of MUSM 223.
This specimen definitely does not correspond to the known anatomy of the dentary of T. natans. Instead, its anatomy is similar to that of T. littoralis, which has several interesting implications that should be developed in the discussion. First, if specimens MPC 704, MPC 705, and SGO PV 1093 correspond to equivalent stratigraphic levels as mentioned in the manuscript, then at least two Thalassocnus taxa coexisted in Bahía Inglesa during the late Miocene. In Peru, to date, no two Thalassocnus species have been discovered at the same stratigraphic level. Thalassonus natans has been recovered from the Tortonian-Messinian (~7.15 Ma), while T. littoralis from the Messinian (6.28-5.6 Ma). Second, because the correspondence between mandible MPC 705 and the only known T. littoralis mandible (MUSM 223) is not precise, the possibility that this is not only intraspecific variation but also that it represents a new species should be discussed. The latter would not be surprising considering the geographical distance of more than 1,600 km separating the Sacaco sites in Peru and Bahía Inglesa in Chile. For MPC 705, it would be ideal to include a comparative figure between the Thalassocnus jaws.

Considering new ages from Pisco Fm, the youngest record of T. yaucensis in the Peruvian coast is ~5.35-4.8. In this new context, if ages of the Horcon and Coquimbo Fm are correct (Late Pliocene), then Thalassocnus youngest records occur in Chile. Conversely, ages of these localities need revision if they were based on the fossils originally found in the Pisco Fm of the Sacaco area.

In summary, evidence suggests that the record from Norte de Bahía de Caldera includes two taxa, one of which is T. natans. The other is morphologically close to T. littoralis (MUSM 223), although there is a possibility that it is a taxon not previously identified in Sacaco, Peru. I suggest including an analysis of linear morphometrics on Thalassonus radii, a comparative figure of Thalassocnus radii from Peru and Chile, and a comparative figure of dentaries from Peru and Chile. The implications of these results should be discussed accordingly. If necessary I can provide images of the dentary of MUSM 223.

- Disparities in marine herbivore guilds across the Pacific. In this manuscript section, the authors stated that ‘size variations among herbivore guilds might have resulted from regional-scale differences in productivity seafloor during the Neogene across the Pacific. The North Pacific had higher productivity and greater availability of food resources (e.g., aquatic vegetation) than the South Pacific during the Neogene, enabling marine herbivorous mammals to reach larger sizes.’ The authors include a chart (figure 16) showing skull length (as indicator of body length) among marine herbivorous across the Pacific vs time (It is not indicated in the figure captions, but I guess triangles represent South Pacific records whereas circles represent records from the North Pacific).

Based on this chart, it is not possible to conclude that northern herbivores were larger than their southern counterparts. It can be inferred that the South Pacific record is basically restricted to the late Miocene-early Pliocene and that the clades that reached the largest size are hardly represented in the South Pacific (Desmostylia and Sirenia; with the exception of a sirenian record in the middle Miocene). It is also observed that desmostylians and sirenians had a wide range of cranial lengths (i.e. body lengths) as mentioned in the manuscript. However, to conclude or at least support whether herbivores from the Northern Hemisphere are larger than those from the Southern Hemisphere is not enough the chart of figure 16, which is a useful preliminary approach. To better visualize the data, it would be also useful to show these data in a comparative histogram. To support the conclusion about size disparity, a possibility is analyzing the data statistically (independent sample testing), restricting the analysis to the time frame with data from both the southern and northern Pacific.

It is true that body size (here evidenced by skull length) is considered an indicator of food resources in the environment, in this case vegetation resources, but bias in the fossil record and phylogenetic signal independent of resources should be discussed since they might be alternative explanations for the data presented. Other factors can also be discussed, such as differences in feeding anatomy among clades, physiology, stage of adaptation to the aquatic environment, etc.

References.
Amson, E., Argot, C., McDonald, H. G., & De Muizon, C. (2015a). Osteology and functional morphology of the forelimb of the marine sloth Thalassocnus (Mammalia, Tardigrada). Journal of Mammalian Evolution, 22, 169-242
Amson, E., Argot, C., McDonald, H. G., & De Muizon, C. (2015b). Osteology and functional morphology of the hind limb of the marine sloth Thalassocnus (Mammalia, Tardigrada). Journal of Mammalian Evolution, 22, 355-419.
Amson, E., Argot, C., McDonald, H. G., & de Muizon, C. (2015c). Osteology and functional morphology of the axial postcranium of the marine sloth Thalassocnus (Mammalia, Tardigrada) with paleobiological implications. Journal of Mammalian Evolution, 22(4), 473-518.
Salas, R., Pujos, F., & de Muizon, C. (2005). Ossified meniscus and cyamo-fabella in some fossil sloths: A morpho-functional interpretation. Geobios, 38(3), 389-394.

Validity of the findings

The manuscript is well-crafted and the new fossil findings provided evidence for provide interesting evidence that helps to understand the diversification of Thalassocnus in the South Pacific. The descriptions are accurate and the identifications only require minor re-evaluations and additions. I consider the conclusions can be more solidly supported by implementing the changes suggested in the previous section, both in taxonomic identification and in addressing differences in herbivory. The manuscript is an important contribution to the emerging understanding of herbivory in the South Pacific and, in this sense, will be an essential starting point for future research.
I endorse the publication of this research in PeerJ after the suggested changes.

Additional comments

Other minor observations:
- Abstract –
Line 36, it says ‘other Thalassocnus.’ The word ‘other’ can be omitted.
the last sentence says: ‘builds on other evidence…’ Which evidence?

- Cooper and Mass, 2018 is cited in line 69 but not included in the reference list.

- Line 91. It says: ‘Here, we report a complete skeleton, including cranial and postcranial remains (MPC 704, 705, 705-A)’. This statement suggests that the three specimens belong to a single individual, which is not the case as has been clearly stated in other parts of the manuscript. Rephrase.

- Line 261. The description of the dentary (MPC 705) lacks details on the predental length vs dental series length ratio, the position of the end of the symphysis and the position of the anterior border of the coronoid process. As mentioned above, this specimen differs in these features from T. natans.

- Line 658, it says: ‘slithly’; it must say: ‘slightly’

- Line 757, Muizon and Domning, 1985 is cited in the text but is not in the reference list. This is the case for several cited references.

- Line 762, Muizon and Domning is incorrectly cited as a reference of a record of a Nanosiren in the Pisco Fm. The material was shown to belong to Thalassonus by Amson et al. 2014.

- Line 1180, In the caption of Figure 16, the triangle symbol shown in the figure is not explained.

In table S1, ages for Thalassocnus yaucensis from Yauca indicate Late Pliocene (3.6-2.588 Ma). These ages need to be corrected following Ochoa et al., 2022, as is stated in sections of the main manuscript.
Table S1, mentions that Lomas del Sauce locality belongs to the Horcon Fm whereas the main text refers it to the Coquimbo Fm, as it is stated in De los Arcos et al., 2017.

---

## Round 0.2 · accepted · Accept

Thank you very much for thoroughly addressing the concerns of the two reviewers. I am glad to see that you agree that the manuscript has been improved. I have now recommended acceptance of the manuscript. Please review the proofs carefully when they are provided. Congratulations on the very nice work.